# Modifiable lifestyle factors and the risk of post-COVID-19 multisystem sequelae, hospitalization, and death

Yunhe Wang [1,9], Binbin Su[2,9], Marta Alcalde-Herraiz[3], Nicola L. Barclay[3], Yaohua Tian [4], Chunxiao Li[5], Nicholas J. Wareham [5], Roger Paredes[6,7], Junqing Xie [3] ✉ & Daniel Prieto-Alhambra [3,8]

Effective prevention strategies for post-COVID complications are crucial for patients, clinicians, and policy makers to mitigate their cumulative burden. This study evaluated the association of modifiable lifestyle factors (smoking, alcohol intake, BMI, physical activity, sedentary time, sleep duration, and dietary habits) with COVID-19 multisystem sequelae, death, and hospitalization in the UK Biobank cohort ($n = 68,896$). A favorable lifestyle (6-10 healthy factors; 46.4%) was associated with a 36% lower risk of multisystem sequelae (HR, 0.64; 95% CI, 0.58-0.69; ARR at 210 days, 7.08%; 95% CI, 5.98-8.09) compared to an unfavorable lifestyle (0-4 factors; 12.3%). Risk reductions spanned all 10 organ systems, including cardiovascular, coagulation, metabolic, gastrointestinal, kidney, mental health, musculoskeletal, respiratory disorders, and fatigue. This beneficial effect was largely attributable to direct lifestyle impacts independent of corresponding pre-infection comorbidities (71% for any sequelae). A favorable lifestyle was also related to the risk of post-COVID death (HR 0.59, 0.52-0.66) and hospitalization (HR 0.78, 0.73-0.84). These associations persisted across acute and post-acute infection phases, irrespective of hospitalization status, vaccination, or SARS-CoV-2 variant. These findings underscore the clinical and public health importance of adhering to a healthy lifestyle in mitigating long-term COVID-19 adverse impacts and enhancing future pandemic preparedness.

COVID-19 cases and deaths have decreased globally, yet the long-term health consequences of SARS-CoV-2 infection, termed as *post-COVID-19 conditions* or *long COVID*, are still being managed as a global public health crisis[1,2]. These conditions or symptoms can involve pulmonary and multiple extrapulmonary organ systems, and may occur or extend beyond the acute infection of varying severity, with significant impact on daily functioning and quality of life[3]. Increased risk and burden of cardiovascular, pulmonary, neuropsychiatric, and metabolic disorders were reported during the 6 to 12 months following SARS-CoV-2 infection[4,5], with persistent risk observed for several diseases up to 2 years[6,7].

[1]Nuffield Department of Population Health, University of Oxford, Oxford, UK. [2]School of Population Medicine and Public Health, Chinese Academy of Medical Sciences/Peking Union Medical College, Beijing, China. [3]Centre for Statistics in Medicine and NIHR Biomedical Research Centre Oxford, NDORMS, University of Oxford, Oxford, UK. [4]School of Public Health, Tongji Medical College, Huazhong University of Science and Technology, Wuhan, China. [5]Medical Research Council Epidemiology Unit, University of Cambridge, Cambridge, UK. [6]Department of Infectious Diseases & irsiCaixa AIDS Research Institute, Hospital Universitari Germans Trias i Pujol, Catalonia, Spain. [7]Center for Global Health and Diseases, Department of Pathology, Case Western Reserve University School of Medicine, Cleveland, OH, US. [8]Department of Medical Informatics, Erasmus Medical Center University, Rotterdam, Netherlands. [9]These authors contributed equally: Yunhe Wang, Binbin Su. ✉e-mail: Junqing.xie@ndorms.ox.ac.uk

Despite long COVID has been characterized, however, evidence-based strategies for its prevention or treatment are not yet available[8,9]. Previous studies on its prevention have mainly focused on vaccination and pharmaceutical approaches, including antivirals (e.g., molnupiravir and nirmatrelvir) and other drugs repurposed for long COVID (e.g., metformin). Increasing evidence suggests that vaccination before infection and use of antivirals during acute phase in selected high-risk patients only partially mediate the risk of long COVID at 6 to 12 months following infection (by 15–51% for vaccination[10–12], by 26% for nirmatrelvir[13], and by 14% for molnupiravir[14]). Several potential drugs for long COVID are still under investigation without yielding reliable results[8,9]. Evidence for the non-pharmaceutical management strategies is also lacking. Effective prevention and intervention strategies are needed to inform patients, clinicians and policy makers, and to reduce the cumulative burden of post-COVID conditions.

Modifiable lifestyle factors such as physical activity and healthy diet are potential targets for the prevention of major non-communicable diseases[15–17], and are associated with lower risk of severe COVID-19 and related mortality[18,19], possibly through protection against inflammation[17,18], autoimmunity[20,21], and clotting abnormality[22,23]. These mechanisms overlap with the hypothesized pathogenesis of long COVID, and other postviral conditions, such as chronic fatigue syndrome[3]. When examined individually, factors such as smoking and obesity, have been reported to be related to increased risk of post-COVID symptoms mainly in hospitalized patients[12]. Nevertheless, association between combinations of multiple lifestyle factors, which are known to interact synergistically[24,25], and risk of COVID-19 sequelae across multiple organ systems remains unclear.

This major knowledge gap should be urgently addressed to inform the prevention and care strategies of long COVID. Based on a large-scale, prospective population-based cohort, we evaluated the relationship between composite healthy lifestyle (including 10 modifiable factors) that predated the pandemic and subsequent risk of COVID-19 sequelae in 10 organ systems, death and hospital admission, while considering the phase of infection (acute or post-acute), severity of infection (tested positive in community/outpatient setting vs inpatient setting), vaccination status (fully vaccinated vs unvaccinated or partially vaccinated), and variants (alpha [B.1.1.7], delta [B.1.617.2], vs omicron [B.1.1.529]), that differ in transmissibility, disease course, and disease severity[26].

## Results

### Baseline characteristics

Out of 472,977 eligible UK Biobank participants, 68,896 participants with a positive SARS-CoV-2 test result between March 1, 2020 and March 1, 2022 were included in the current study. The demographic and health characteristics of the eligible participants, and participants with COVID-19 overall and by healthy lifestyle category are provided in Table 1. Of the COVID-19 cohort, the mean (SD) age was 66.6 (8.4) years, 53.4% were male and 82.1% were White. For composite healthy lifestyle prior to the infection, 12.3% followed an unfavorable lifestyle, 41.3% followed an intermediate lifestyle, and 46.4% followed a favorable lifestyle. The median [IQR] number of healthy lifestyle factors participants engaged in was 7 [6–8]. For prespecified COVID-19 sequalae, 5.5% and 7.8% had sequelae in at least one organ system during the acute and post-acute phase of infection, respectively.

### Risk of multisystem sequelae

Overall, the risk of multisystem COVID-19 sequelae decreased monotonically across healthy lifestyle categories during both the acute and post-acute phases of infection. Compared with those with an unfavorable lifestyle, participants with an intermediate (HR, 0.80; 95% CI, 0.74–0.87; ARR at 210 days, 3.89%; 95% CI, 2.56–5.12) and favorable (HR, 0.64; 95% CI, 0.58–0.69; ARR at 210 days, 7.08%; 95% CI, 5.98–8.09) lifestyle were at significantly lower risk of multisystem sequelae of COVID-19 (Fig. 1), with similar trends observed in both the acute and post-acute phases of COVID-19 (Fig. 2). The number of healthy lifestyle factors (range, 0–10) was associated with risk of sequelae in a dose-dependent manner (Fig. 1).

The inverse associations with multisystem sequelae were largely attributable to the direct protective effect of a healthy lifestyle (proportion of direct effect on any sequela: 71%), with proportion of direct effect ranging from 44% to 93% across organ systems (Fig. 3a). Pre-infection medical conditions were associated with substantially increased risk of COVID-19 sequelae, particularly history of cardiovascular diseases, diabetes and mental disorders (Fig. 3b). Number of participants with medical conditions between baseline and infection is provided in Supplementary Table 4.

For individual components of the healthy lifestyle, each of the 10 studied behavioral and dietary factors was associated with a lower or non-differential risk of sequelae, with smoking, physical activity, obesity, and sleep duration contributing most (Fig. 4).

### Risk of death and hospitalization

Adherence to a healthy lifestyle was associated with lower risk of death and hospitalization following COVID-19 during both the acute and post-acute phases of infection. Compared with those with an unfavorable lifestyle, participants with a favorable lifestyle were at significantly lower risk of death (HR, 0.59; 95% CI, 0.52–0.66; ARR at 210 days, 1.99%; 95% CI, 1.61–2.32) and hospitalization (HR, 0.78; 95% CI, 0.73–0.84; ARR at 210 days, 6.14%; 95% CI, 4.48–7.68) following COVID-19 (Fig. 1), with similar trend observed in both the acute and post-acute phases (Fig. 2). Each of 10 lifestyle factors was associated with a lower or non-differential risk of death and hospitalization (Fig. 4).

### Risk of system-specific sequelae

Compared with those following an unfavorable lifestyle, participants with a favorable lifestyle had a significantly lower risk of sequelae in all 10 organ systems examined, including cardiovascular, coagulation and hematologic, metabolic and endocrine, gastrointestinal, kidney, mental health, musculoskeletal, neurologic, and respiratory disorders, as well as general symptoms of fatigue and malaise, with overall HRs ranging from 0.38 to 0.76 (Fig. 1). The associations with intermediate lifestyle were consistently in the same protective direction across system-specific sequelae (Fig. 1). Similar trend were observed in both the acute and post-acute phases (Fig. 2).

### Risk of outcomes by subgroups

The inverse associations between healthy lifestyle and risk of multisystem sequelae, death, and hospitalization held across the different subgroups of clinical interest, including those by age, sex, and ethnicity, vaccine status, test setting, and variants of infection (Table 2). The reduced risk of outcomes was observed in participants who received two doses of vaccine (breakthrough infection) and those who were unvaccinated or partially 1-dose vaccinated (non-breakthrough infection). The reduced risk was evident in participants tested positive in inpatient setting and in those tested positive in community/outpatient settings. The reduced risk was consistently observed across predominant variants of SARS-CoV-2 infection during study period, including wildtype, Alpha, Delta, and Omicron BA.1. Notably, a composite healthy lifestyle was associated with decreased risk of outcomes following infection of Omicron variant, which remains currently dominant variant worldwide. No significant interaction was observed in any of the subgroups across outcomes, except for age, ethnicity, and vaccination. The observed association between a favorable lifestyle and a reduced risk of outcomes was more evident for people aged < 65, in white subjects, and in those fully vaccinated (for mortality only).

**Table 1 | Baseline characteristics of eligible participants, and COVID-19 cohort overall and by lifestyle category**

| Characteristics | All eligible participants[a] | Participants with COVID-19 | | | |
|---|---|---|---|---|---|
| | | Overall | By composite lifestyle category[b] | | |
| | | | Unfavorable | Intermediate | Favorable |
| No. (%) | 472977 | 68896 | 8476 (12.3) | 28457 (41.3) | 31963 (46.4) |
| Age, mean (SD) | 68.1 (8.1) | 66.6 (8.4) | 66.6 (8.3) | 66.8 (8.4) | 66.5 (8.4) |
| Sex, No. (%) | | | | | |
| Male | 261436 (55.3) | 36798 (53.4) | 3206 (37.8) | 14207 (49.9) | 19385 (60.6) |
| Female | 211540 (44.7) | 32098 (46.6) | 5270 (62.2) | 14250 (50.1) | 12578 (39.4) |
| Ethnicity, No. (%) | | | | | |
| White | 382010 (80.8) | 56566 (82.1) | 6842 (80.7) | 23247 (81.7) | 26477 (82.8) |
| Other ethnic groups[c] | 90967 (19.2) | 12330 (17.9) | 1634 (19.3) | 5210 (18.3) | 5486 (17.2) |
| Education categories,[d] No. (%) | | | | | |
| I | 85686 (18.1) | 11398 (16.5) | 1946 (23.0) | 5140 (18.1) | 4312 (13.5) |
| II | 125549 (26.5) | 20644 (30.0) | 2826 (33.3) | 8670 (30.5) | 9148 (28.6) |
| III | 52738 (11.2) | 7748 (11.2) | 869 (10.3) | 3112 (10.9) | 3767 (11.8) |
| IV | 24148 (5.1) | 3205 (4.7) | 364 (4.3) | 1340 (4.7) | 1501 (4.7) |
| V | 184856 (39.1) | 25901 (37.6) | 2471 (29.2) | 10195 (35.8) | 13235 (41.4) |
| Index of multiple deprivation,[e] mean (SD) | 17.5 (12.9) | 18.6 (13.6) | 21.9 (15.3) | 19.4 (13.9) | 17.1 (12.5) |
| Individual lifestyle factors, No. (%) | | | | | |
| Smoking | | | | | |
| Never or past | 425810 (90.0) | 62066 (90.1) | 6043 (71.3) | 25126 (88.3) | 30897 (96.7) |
| Current | 47167 (10.0) | 6830 (9.9) | 2433 (28.7) | 3331 (11.7) | 1066 (3.3) |
| Alcohol consumption | | | | | |
| ≤ 4 times a week | 377958 (79.9) | 56248 (81.6) | 5526 (65.2) | 21944 (77.1) | 28778 (90.0) |
| Daily or almost daily | 95019 (20.1) | 12648 (18.4) | 2950 (34.8) | 6513 (22.9) | 3185 (10.0) |
| BMI | | | | | |
| < 30 kg/m$^2$ | 359704 (76.1) | 51001 (74.0) | 3592 (42.4) | 19011 (66.8) | 28398 (88.8) |
| ≥ 30 kg/m$^2$ | 113273 (23.9) | 17895 (26.0) | 4884 (57.6) | 9446 (33.2) | 3565 (11.2) |
| Physical activity | | | | | |
| ≥ 150 min/week of MIPA or ≥75 min/ week of VIPA | 398880 (84.3) | 57930 (84.1) | 4913 (58.0) | 22857 (80.3) | 30160 (94.4) |
| <150 min/week of MIPA and <75 min/ week of VIPA | 74097 (15.7) | 10966 (15.9) | 3563 (42.0) | 5600 (19.7) | 1803 (5.6) |
| TV viewing/sedentary time | | | | | |
| < 4 h/day | 339946 (71.9) | 50500 (73.3) | 3388 (40.0) | 18819 (66.1) | 28293 (88.5) |
| ≥ 4 h/day | 133031 (28.1) | 18396 (26.7) | 5088 (60.0) | 9638 (33.9) | 3670 (11.5) |
| Sleep duration | | | | | |
| 7–9 h/day | 349546 (73.9) | 50777 (73.7) | 3718 (43.9) | 18982 (66.7) | 28077 (87.8) |
| <7 or >9 h/day | 123431 (26.1) | 18119 (26.3) | 4758 (56.1) | 9475 (33.3) | 3886 (12.2) |
| Fruit and vegetable intake | | | | | |
| ≥ 400 g/day | 377154 (79.7) | 53635 (77.8) | 3699 (43.6) | 20696 (72.7) | 29240 (91.5) |
| < 400 g/ day | 95823 (20.3) | 15261 (22.2) | 4777 (56.4) | 7761 (27.3) | 2723 (8.5) |
| Oily fish intake | | | | | |
| ≥ 1 portion/week | 265642 (56.2) | 35996 (52.2) | 1727 (20.4) | 11504 (40.4) | 22765 (71.2) |
| <1 portion/week | 207335 (43.8) | 32900 (47.8) | 6749 (79.6) | 16953 (59.6) | 9198 (28.8) |
| Red meat intake | | | | | |
| ≤ 3 portion/week | 216206 (45.7) | 34003 (49.4) | 3375 (39.8) | 12650 (44.5) | 17978 (56.2) |
| > 3 portion/week | 256771 (54.3) | 34893 (50.6) | 5101 (60.2) | 15807 (55.5) | 13985 (43.8) |
| Processed meat intake | | | | | |
| ≤ 1 portion/week | 328721 (69.5) | 46295 (67.2) | 2536 (29.9) | 16352 (57.5) | 27407 (85.7) |
| > 1 portion/week | 144256 (30.5) | 22601 (32.8) | 5940 (70.1) | 12105 (42.5) | 4556 (14.3) |

BMI body mass index (calculated as weight in kilograms divided by height in meters squared), MIPA moderate-intensity physical activity, VIPA vigorous-intensity physical activity.

[a]All eligible participants included were those who were alive when the study began (March 1, 2020) in UK Biobank.

[b]Participants were classified into three lifestyle categories: favorable (0–4 unhealthy lifestyle factors), intermediate (5–6), and unfavorable (7–10), based on the composite score of 10 modifiable lifestyle factors including smoking, alcohol consumption, BMI, physical activity, TV watching/sedentary time, sleep duration, intake of fruit and vegetables, oily fish intake, red meat intake, and processed meat intake.

[c]Participants with missing values (missing or unknown) in ethnicity were classified as "other ethnic groups": 65682 (13.9%) for all eligible participants and 8128 (11.8%) for participants with COVID-19.

[d]Education levels were mapped to the international standard for the classification of education. Category I includes self-reported "None of the above" and "Prefer not to answer", II includes "CSEs or equivalent" and "O levels/GCSEs or equivalent", III includes "A levels/AS levels or equivalent", IV includes "Other professional qualifications e.g., nursing, teaching", and V includes "NVQ or HND or HNC or equivalent" and "College or University degree."

[e]A summary measure of crime, education, employment, health, housing, income, and living environment, with a high score indicating higher levels of deprivation.

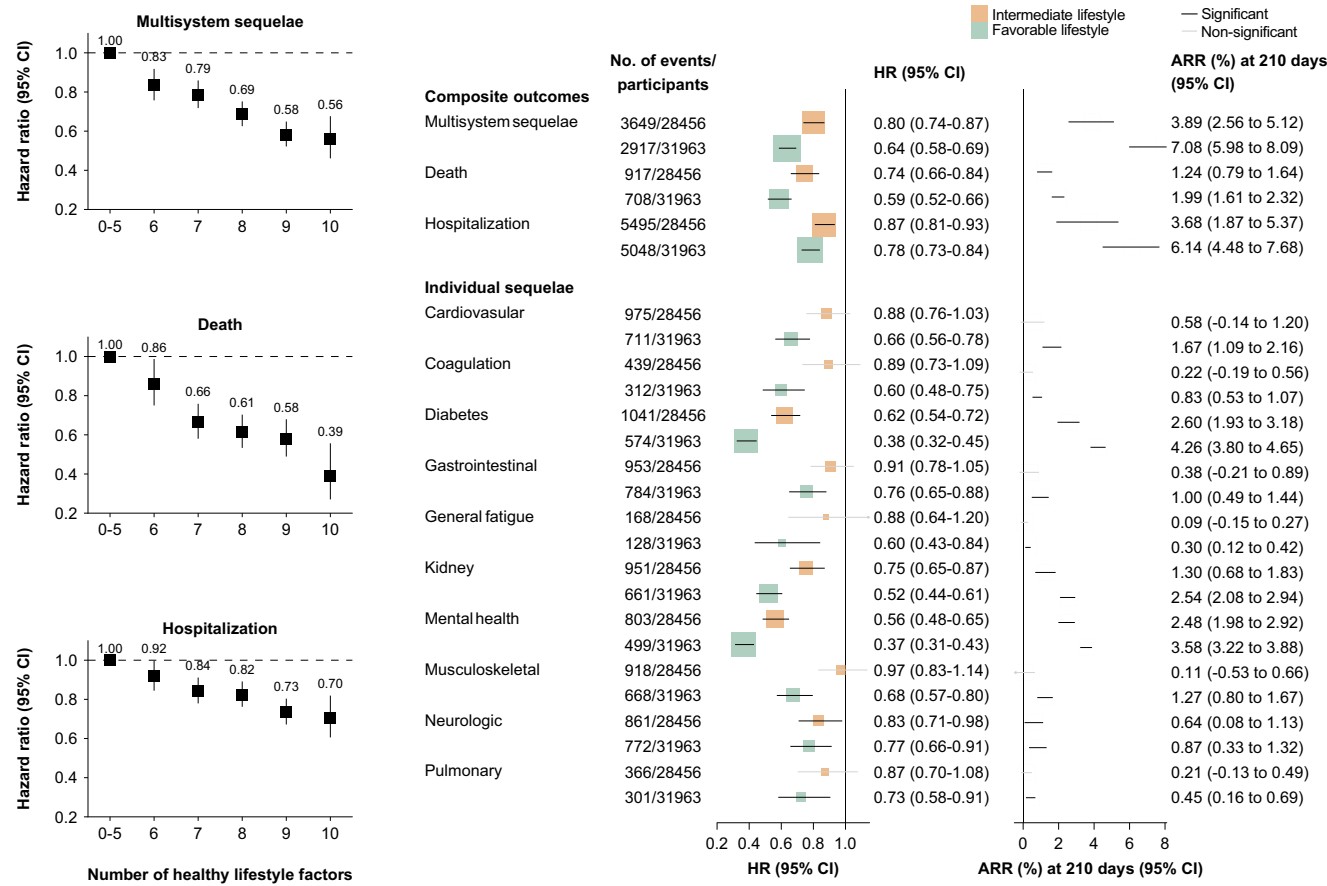

**Fig. 1 | Association of healthy lifestyle with multisystem sequelae of COVID-19, death, and hospital admissions, during overall phase of SARS-CoV-2 infection.** Healthy lifestyle (composite or number) and risk of multisystem sequelae (composite or by organ systems), death, and hospitalization during the overall phase (0–210 days) of SARS-CoV-2 infection. Adjusted HRs and 95% CI are presented for composite/individual multisystem sequelae, death, and hospitalization. Absolute risk reduction (ARR) per 100 persons at 210 days and 95% CIs were calculated. Solid squares represent HRs with the area inversely proportional to the variance of the log HR. Hollow square represents ARR. The horizontal lines indicate 95% CIs, with black line representing statistically significant results and gray line representing non-significant results. Intermediate lifestyle category are in orange, favorable lifestyle category in green.

## Sensitivity analyses

A similar pattern of associations was observed in multiple sensitivity analyses, including assigning weight to individual sequela and using zero-inflated poisson regression to estimate the association, excluding participants with history of related outcomes in the past two years rather than one year, defining post-acute outcomes 90 days after infection rather than 30 days, and restricting the identification of outcomes to the first three ICD diagnoses (Supplementary Table 5). Stronger associations were identified after accounting for potential misclassification of lifestyle factors, suggesting that the observed associations may have been underestimated (Supplementary Table 5).

## Risk of outcomes in the uninfected group

The associations between healthy lifestyle and risk of three predefined main outcomes were largely similar among participants with SARS-CoV-2 infection and those with no evidence of infection during the overall or 30–210 days of follow-up (Fig. 5 and Supplementary Table 6). However, the inverse associations between healthy lifestyle and long COVID-associated complications and hospitalization were more evident during 0-30 days in the infected group (acute phase) compared to those in the uninfected group; whereas the association with death was stronger in the uninfected group (Supplementary Table 6).

## Discussion

Based on a large, prospective population-based cohort, this study provides a comprehensive assessment of the health effects of multiple lifestyle factors on a systematic range of disease outcomes following COVID-19. Adherence to a healthy lifestyle prior to infection was associated with significantly lower risk of COVID-19 multisystem sequelae, death, and hospital admission, during both the acute and post-acute phases of SARS-CoV-2 infection. The reduced risk was evident across 10 prespecified organ systems, including cardiovascular, coagulation and hematologic, metabolic and endocrine, gastrointestinal, kidney, mental health, musculoskeletal, neurologic, and respiratory disorders, as well as general symptoms of fatigue and malaise. The reduced risk of multisystem sequelae associated with a healthy lifestyle was consistently observed across participants, regardless of their vaccination status (unvaccinated/partially vaccinated or fully vaccinated), disease severity (testing positive in community/outpatient settings or inpatient settings), and major SARS-CoV-2 variants, including Omicron variants, whose subvariants are currently dominant. Moreover, the benefits of healthy lifestyle on sequelae were largely independent of pathways related to pre-existing relevant disease conditions. Overall, the findings suggest that adherence to a healthy lifestyle prior to infection was consistently and directly associated with reduced risk of adverse health outcomes following COVID-19.

We found that a favorable lifestyle, in comparison with an unhealthy one, was associated with a 36% lower risk of multisystem sequelae, a 41% lower risk of death, and a 22% lower risk of hospitalization, which corresponded to an absolute risk reduction of 7.08, 1.99, and 6.14 fewer cases per 100 people at 210 days after infection. This

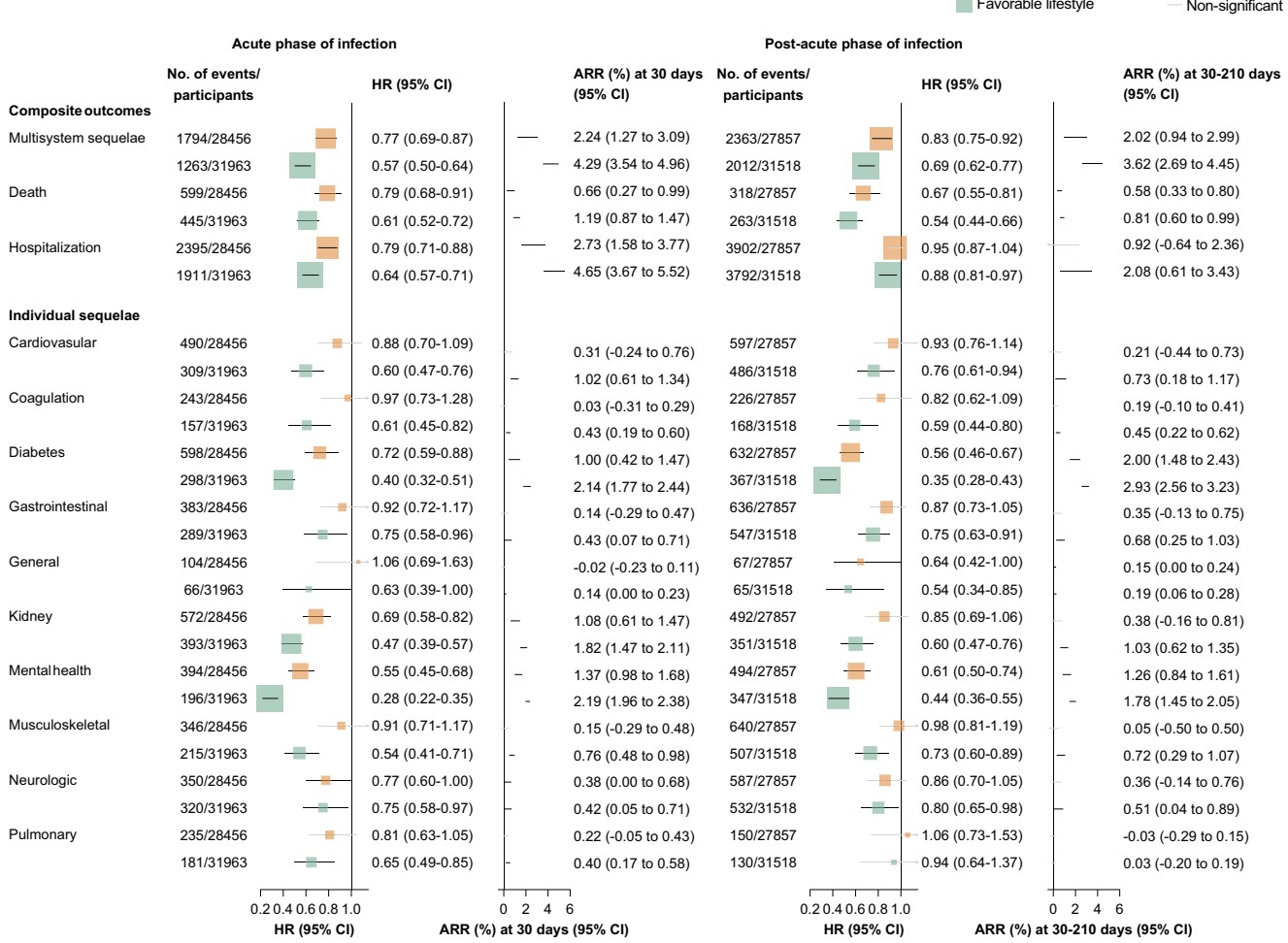

**Fig. 2 | Association of healthy lifestyle with multisystem sequelae of COVID-19, death, and hospital admissions, during acute and post-acute phases of SARS-CoV-2 infection.** Composite healthy lifestyle and risk of multisystem sequelae, death, and hospitalization during the acute phase (first 30 days) and post-acute (30–210 days) phases of SARS-CoV-2 infection. Adjusted HRs and 95% CIs are presented for composite/individual multisystem sequelae, death, and hospitalization. Absolute risk reduction (ARR) per 100 persons at 30 days and 30–210 days and 95% CIs were calculated. Solid square represents HRs with the area inversely proportional to the variance of the log HR. Hollow square represents ARR. The horizontal lines indicate 95% CIs, with black line representing statistically significant results and gray line representing non-significant results. Intermediate lifestyle category are in orange, favorable lifestyle category in green.

association was even larger than those observed in previous studies of pharmaceutical interventions in non-hospitalized patients, which reported a 14% risk reduction in post-acute sequelae at 180 days for vaccination before infection, and 14% and 26% risk reductions at 180 days for the use of molnupiravir and nirmatrelvir during acute phase of infection, respectively[10,13,14]. It is important to note that participants with breakthrough infection were still at risk of sequelae compared with those without infection[10]. In addition, only selected patients at risk of progression to severe COVID-19 are qualified for antivirals during the acute infection[13,14], and their benefit-risk profile in wider population with milder infection, or when used during the post-acute stage, remains unclear. These previous findings highlighted the restricted scope of currently available therapies and limited efficacy of vaccination in preventing long COVID[10,13,14,27]. Our results are consistent with a cross-sectional study of 1981 women suggesting an inverse association between composite healthy lifestyle (mainly driven by BMI and sleep duration) and self-reported symptoms following infection of non-Omicron variants[28]. However, outcomes purely based on self-report symptoms are less clinically relevant and the inclusion of only women may limit the generalization of findings to other populations and settings.

The mechanisms underlying the benefit of adhering to a healthy lifestyle for the alleviation of sequelae are likely multifaceted. Previous research has established causal links between several individual lifestyle factors, such as smoking, obesity, and physical inactivity, and increased susceptibility and severity in relation to COVID-19[29]. Smoking and high BMI were also risk factors for long COVID symptoms mainly in hospitalized patients[12]. Indeed, we observed that these factors and additionally sleep duration, and sedentary behavior were significant contributors to the reduced risk of sequelae. Also, we found that the inverse associations with the risk of long COVID-related complications and hospitalization were more pronounced in the infected group during the first 30 days than in the uninfected group, suggesting the particular benefits of a healthy lifestyle in preventing acute outcomes that could be directly caused by viral infection. However, these pathways are unlikely to fully explain the protective effects on post-infection adverse outcomes conferred by a healthy lifestyle. In our study, all participants had a confirmed infection, and the protective associations persisted even among those who were hospitalized. In addition, it has been suggested that individuals with an unhealthy lifestyle are more likely to have prevalent chronic conditions, such as cardiovascular diseases and diabetes—which are strong

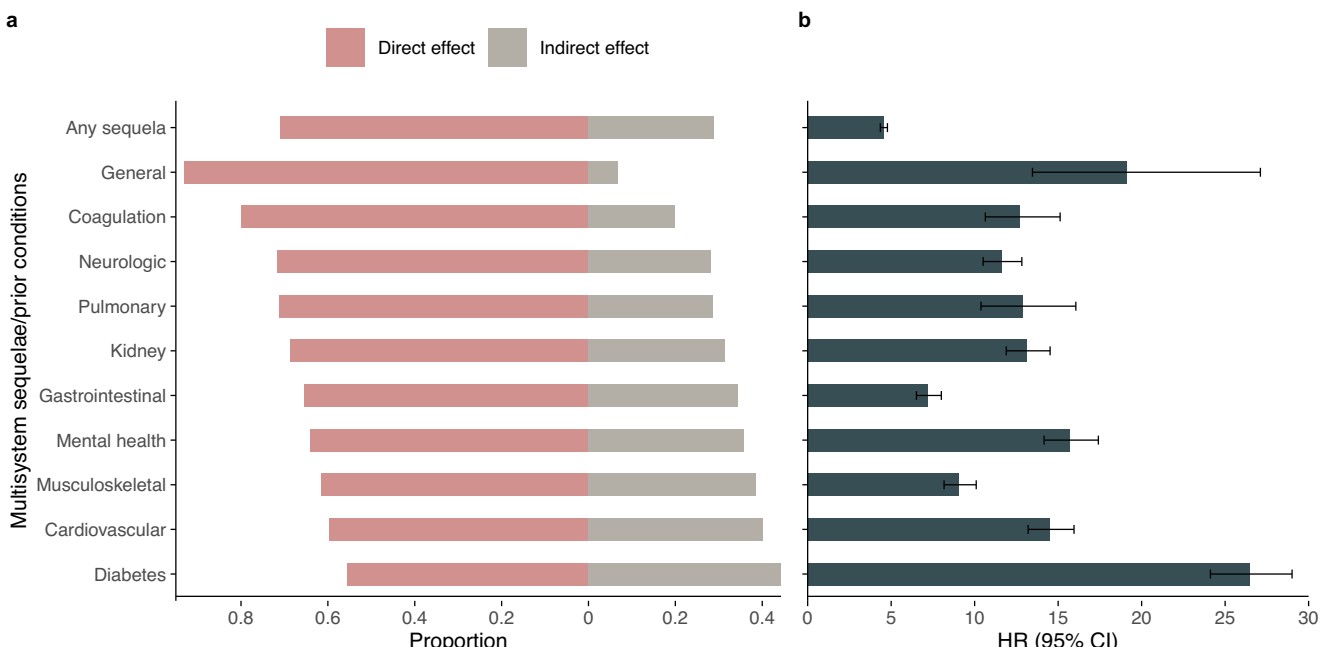

**Fig. 3 | Direct and indirect effects of healthy lifestyle, and association of pre-infection medical conditions with multisystem sequelae of COVID-19.**
**a** Proportion of the direct and indirect effect of a healthy lifestyle on multisystem sequelae (intermediate/favorable vs unfavorable lifestyle). Direct associations were accounted for pre-infection medical conditions (mediator), identified as any relevant event recorded between baseline measurement and infection date.
**b** Association of corresponding pre-infection medical conditions with the risk of

sequelae following SARS-CoV-2 infection. Outcomes were ascertained 0–210 days after SARS-CoV-2 infection. The horizontal bars indicate HR and lines indicate 95% CIs. The sample size was 68,892. 7975 incident events for any sequela, 354 for general fatigue, 923 for coagulation diseases, 1938 for neurologic diseases, 800 for pulmonary diseases, 2023 for Kidney diseases, 2064 for gastrointestinal diseases, 1739 for mental disorders, 1895 for musculoskeletal diseases, 2077 for cardiovascular diseases, and 2152 for diabetes.

risk factors for severe COVID-19—and are therefore more vulnerable to post-acute complications. Through mediation analysis, our study supported this hypothesis and, for the first time, further demonstrated that the healthy lifestyle's direct protection accounts for the majority of the overall associations with COVID-19 sequelae. Notably, varying proportions of indirect effects from healthy lifestyle were observed depending on the specific sequela of interest. For example, pre-existing diabetes had the strongest association with post-infection diabetes sequela, therefore lifestyle is more likely to confer its protection indirectly through this pathway. In contrast, thrombotic events are generally more acute and transient, making it less likely for healthy lifestyle to confer indirect protection. Biologically, the overlapping mechanisms between unhealthy lifestyle and viral infection and post-infection conditions may also be involved. Favorable lifestyle factors, such as physical activity and healthy diet, confer health benefits including protection against inflammation[17,18], autoimmunity[21,30], and clotting abnormality[23], which are implicated in the potential pathogenesis of long COVID[3].

Although our findings align with previous evidence on the broader benefits of healthy lifestyle on chronic disease prevention and life expectancy[15–17], this potentially beneficial effect should not be interpreted as changing behaviors around the time of acute infection or during post-acute infection. The current practical guide recommends that patients without long COVID should gradually and safely return to pre-infection physical activity when appropriate, although direct evidence is lacking[31]. Previous study also characterized long COVID as a multifactorial condition determined by pathogen, host response, external pandemic-associated factors, and supported a multidisciplinary treatment including both pharmacological and rehabilitation approaches, but also social and welfare support to promote healthy lifestyle habits[32,33]. Future research is warranted to assess the effect of composite lifestyle interventions in prevention of long

COVID or alleviating associated symptoms among patients with long COVID. Adherence to a healthy lifestyle, in combination with vaccination and, if necessary, potential medications, may be a viable and practical approach to further reduce the long-term health consequences of SARS-CoV-2 infection. These strategies hold significant public health and scientific importance in mitigating the overall burden of post-COVID complications and enhancing preparedness for future pandemics.

This study has several limitations. First, the UK Biobank participants are likely to be older than the general population of the UK and are mostly of European ancestry, which may limit the generalizability of study findings to the younger population and other ethnic groups. Second, the majority of participants (87%) were classified as intermediate or favorable lifestyle category, suggesting that the study population appears to be healthier than the wider general population. However, the exact distribution of lifestyle categories based on the similar 10 modifiable factors in another population remains unknown. A previous study in Australia (*n* = 231,048)[34] reported 68% of participants had ≤1 unhealthy lifestyle factor out of 7 factors assessed (smoking, alcohol consumption, physical activity, sitting time, sleeping duration, and diet), which partly in line with our results. Assuming participants are healthier than the UK general population, the absolute risk estimates such as ARR should be interpreted with caution. Nevertheless, the relative associations of risk factors with disease outcomes in the UK Biobank were tested to be generalizable and comparable to those from other representative cohort of general population[35,36]. In addition, this high proportion could also suggest potential misreporting bias in self-reported lifestyle data. However, healthy reporting bias may be more common in socioeconomically deprived individuals[37] and tends to bias any genuine association towards null. Self-reported lifestyle data, such as sleep duration[38] and physical activity[39], have been shown to be highly correlated with

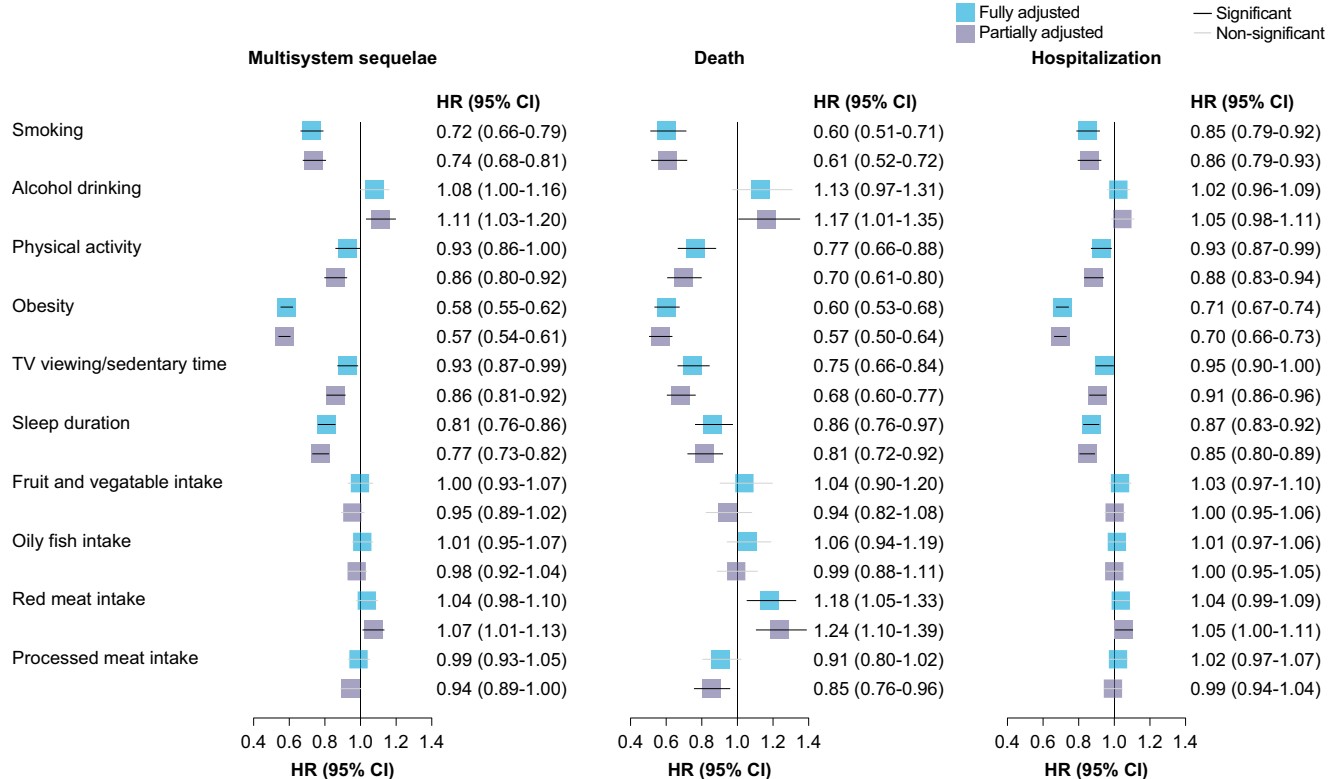

**Fig. 4 | Association of individual healthy lifestyle with multisystem sequelae, death, and hospitalization.** Blue square represents risk estimates from models fully adjusted for age, sex, education level, ethnicity, IMD, and mutually for all lifestyle factors. The purple square represents risk estimates from models partially adjusted for age, sex, education level, ethnicity, and IMD. The horizontal lines indicate 95% CIs, with black line representing statistically significant results and the gray line representing non-significant results. The sample sizes were 60,561 for any sequela (4792 events), 55,106 for hospitalization (6958 events), and 68,887 for death (1203 events). The HR for each lifestyle factor was calculated by comparing the healthy category with the unhealthy category (e.g., past or never smoker versus current smoker).

accelerometer-derived measures in UK Biobank. Third, residual confounding and reverse causality cannot be ruled out in this observational study. Fourth, we assumed the baseline lifestyle unchanged over years until the time of infection, which may be subject to exposure misclassification and underestimated any genuine associations. However, reassuringly, there was high consistency of lifestyle measures between cohort baseline and repetitive interviews after a median of 8 years, and consistent associations were observed after accounting for potential changes in lifestyle factors. Fifth, as sequelae outcomes were based on inpatient records, milder long COVID symptoms were less likely to be detected. Sixth, given the potential non-linear effects of lifestyle factors, such as alcohol consumption, caution is warranted when interpreting associations between binarized lifestyle factors and outcomes. The health effects and recommended targets of several individual lifestyle factors, such as alcohol consumption and red meat intake, are inconsistent in previous epidemiological studies and guidelines[40,41], and may potentially vary by disease outcome of interest. Findings on the association of such individual factors with post-infection complications should be considered in the wider context of chronic disease, whether directly related to infection or not. Seventh, it's important to acknowledge that some participants classified as uninfected may have had undiagnosed or untested COVID-19. However, by linking participants to official national databases for COVID-19 testing and hospitalization, the likelihood of misclassifying infected and uninfected participants was minimized. Finally, despite we included a range of sequelae across organ systems, it is difficult to link these outcomes directly to the infection, especially given the lack of consensus standard for diagnosis of long COVID. This limitation is applied to all related studies in the field[42–50]. Nevertheless, the sequelae prespecified were most relevant to long COVID based on prior evidence, with increased risk and burden consistently reported beyond the acute phase in the currently largest electronic dataset[42–48], the same UK Biobank[50], and other nationwide cohorts[49,50].

Adherence to a healthy lifestyle that predated the pandemic was associated with substantially lower risk of sequelae across organ systems, death, and hospitalization following COVID-19, regardless of phases of infection, vaccination status, test setting, and SARS-CoV-2 variants, and independent of relevant comorbidities. These findings suggest the benefit of population adhering to a healthy lifestyle to reduce the potential long-term adverse health consequences of COVID-19.

## Methods

### Data sources and study cohorts

UK Biobank is a large-scale population-based prospective cohort study with deep phenotyping and genomic data, as detailed elsewhere[51]. Briefly, between 2006 and 2010, over 500,000 individuals aged 40–69 years were recruited from 22 assessment centers across the United Kingdom at baseline, with collection of socio-demographic, lifestyle and health-related factors, a range of physical measures, and blood samples[51]. Follow-up information is obtained by linking health and medical records, including national primary and secondary care, disease and mortality registries[52], with validated reliability, accuracy and completeness[53]. To identify cases of SARS-CoV-2 infection, polymerase chain reaction (PCR)-based test results were obtained by linking all participants to the Public Health England's Second Generation Surveillance System, with dates of specimen collection and healthcare settings of testing[54]. Outbreak dynamics were validated to be broadly similar between UK Biobank participants and the general population of England[54].

**Table 2 | Association of composite healthy lifestyle with multisystem sequelae, death, and hospitalization following SARS-CoV-2 infection in key clinical subgroups**

| | Multisystem sequelae | | Death | | Hospitalization | |
|---|---|---|---|---|---|---|
| | HR (95% CI) | P for interaction | HR (95% CI) | P for interaction | HR (95% CI) | P for interaction |
| **Age, y** | | | | | | |
| < 65 | 0.74 (0.64–0.85) | 0.08 | 0.58 (0.39–0.85) | 0.06 | 0.86 (0.76–0.96) | 0.47 |
| | 0.51 (0.44–0.59) | <0.001 | 0.39 (0.25–0.61) | 0.01 | 0.69 (0.61–0.78) | 0.03 |
| ≥ 65 | 0.84 (0.76–0.93) | | 0.77 (0.68–0.88) | | 0.88 (0.81–0.97) | |
| | 0.70 (0.64–0.78) | | 0.61 (0.54–0.70) | | 0.83 (0.76–0.91) | |
| **Sex** | | | | | | |
| Female | 0.78 (0.69–0.89) | 0.30 | 0.63 (0.51–0.78) | 0.04 | 0.82 (0.73–0.92) | 0.42 |
| | 0.58 (0.51–0.67) | 0.02 | 0.52 (0.42–0.64) | 0.09 | 0.73 (0.65–0.81) | 0.14 |
| Male | 0.80 (0.72–0.89) | | 0.81 (0.70–0.93) | | 0.90 (0.82–0.99) | |
| | 0.69 (0.61–0.77) | | 0.62 (0.53–0.73) | | 0.83 (0.75–0.91) | |
| **Ethnicity** | | | | | | |
| White | 0.75 (0.69–0.82) | 0.01 | 0.74 (0.65–0.85) | 0.67 | 0.84 (0.78–0.90) | 00.04 |
| | 0.59 (0.54–0.64) | <0.001 | 0.58 (0.50–0.67) | 0.36 | 0.73 (0.68–0.79) | <0.001 |
| Other ethnic groups | 1.10 (0.86–1.42) | | 0.78 (0.61–1.01) | | 1.01 (0.81–1.25) | |
| | 0.93 (0.72–1.21) | | 0.65 (0.50–0.85) | | 1.02 (0.82–1.27) | |
| **Vaccine status**[b] | | | | | | |
| No or partial vaccination | 0.88 (0.78–0.99) | 0.13 | 0.85 (0.74–0.98) | 0.02 | 0.89 (0.80–0.99) | 0.63 |
| | 0.72 (0.63–0.81) | 0.21 | 0.69 (0.60–0.80) | 0.04 | 0.78 (0.70–0.87) | 0.19 |
| Full vaccination | 0.75 (0.67–0.84) | | 0.62 (0.49–0.79) | | 0.86 (0.78–0.95) | |
| | 0.60 (0.53–0.67) | | 0.53 (0.41–0.68) | | 0.79 (0.72–0.87) | |
| **Test setting** | | | | | | |
| Inpatient | 0.80 (0.73–0.88) | 0.91 | 0.80 (0.67–0.96) | 0.87 | 0.89 (0.82--0.97) | 0.07 |
| | 0.64 (0.58–0.70) | 0.42 | 0.64 (0.53–0.77) | 0.33 | 0.80 (0.74–0.87) | 0.13 |
| Community/outpatient | 0.83 (0.70–0.99) | | 0.78 (0.66–0.92) | | 0.81 (0.68–0.95) | |
| | 0.75 (0.63–0.90) | | 0.69 (0.58–0.83) | | 0.81 (0.68–0.96) | |
| **Variants**[a] | | | | | | |
| Wildtype, alpha, and delta | 0.76 (0.69–0.84) | 0.07 | 0.77 (0.68–0.88) | 0.56 | 0.80 (0.73–0.87) | 0.01 |
| | 0.63 (0.57–0.70) | 0.27 | 0.60 (0.53–0.69) | 0.44 | 0.76 (0.69–0.83) | 0.06 |
| Omicron BA.1 | 0.94 (0.80–1.09) | | 0.70 (0.51–0.95) | | 1.05 (0.92–1.20) | |
| | 0.72 (0.61–0.84) | | 0.69 (0.51–0.95) | | 0.89 (0.78–1.01) | |

The first row of each category of the subgroup refers to intermediate vs unfavorable lifestyle, and the second row refers to favorable vs unfavorable lifestyle. Risk estimates were derived from the Cox model including covariates of age, sex, education level, index of multiple deprivations, ethnicity, in addition to a healthy lifestyle and a multiplicative interaction term of the healthy lifestyle with the stratification variable of interest.

P values are two-sided and no statistical adjustment is made for multiple comparisons across subgroups.

[a]The calendar periods of dominant variants in UK were based on pandemic data from the Office for National Statistics: wildtype (March 1, 2020–December 7, 2020), alpha [B.1.1.7] (December 8, 2020–May 17, 2021), delta [B.1.617.2] (May 18, 2021–December 13, 2021), Omicron [B.1.1.529 BA.1] (December 14, 2021–March 1, 2022).

[b]Subgroup analysis by vaccine status was conducted in participants from England, as vaccination data for Scotland and Wales were not available.

In this study, we included participants who were alive by March 1, 2020 and had a positive SARS-CoV-2 PCR test result between March 1, 2020 (date of the first recorded case in the UK Biobank), and March 1, 2022, with the date of first infection considered as index date ($T_0$). For those diagnosed with COVID-19 in hospital, we defined $T_0$ as the date of hospital admission minus a random number of 7 days. The major prevalent variants during the study period included wildtype, Alpha (B.1.1.7), Delta (B.1.617.2), and Omicron (B.1.1.529 BA.1). The calendar periods of dominant variants in the UK were based on pandemic data from the Office for National Statistics[26]. Participants with missing data on study exposures at baseline were excluded. We addressed missing data on covariates using the following approaches: (1) participants with missing values in age and sex (< 0.1%) were excluded. (2) participants with missing values in ethnicity were classified as "other ethnic groups". (3) participants with missing values in education level (0.9%) were classified as "category I", which includes "none of the above" and "prefer not to answer". (4) missing values in IMD (13.8%) were imputed with the mean value of the entire UK Biobank cohort. All participants included in this study provided written informed consent at

recruitment. This study followed the Strengthening the Reporting of Observational Studies in Epidemiology (STROBE) reporting guidelines and received ethical approval from the UKBB ethics advisory committee. Study design, cohort construction, and timeline are provided in Supplementary Fig. 1. All participants provided written informed consent at the UK Biobank cohort recruitment. This study received ethical approval from UK Biobank Ethics Advisory Committee (EAC) and was performed under the application of 65397.

**Lifestyle factors**
Ten prespecified potentially modifiable lifestyle factors were assessed, including smoking, alcohol consumption, body mass index (BMI), physical activity, sedentary time, sleep duration, intake of fruit and vegetable, intake of oily fish, intake of red meat, and intake of processed meat. Selection and categorization of lifestyle factors was based on literature review, previous knowledge, and UK national health service guidelines[55,56]. Multiple lifestyle factors were measured by validated questionnaire for all participants at baseline recruitment. Detailed definitions on measurement and

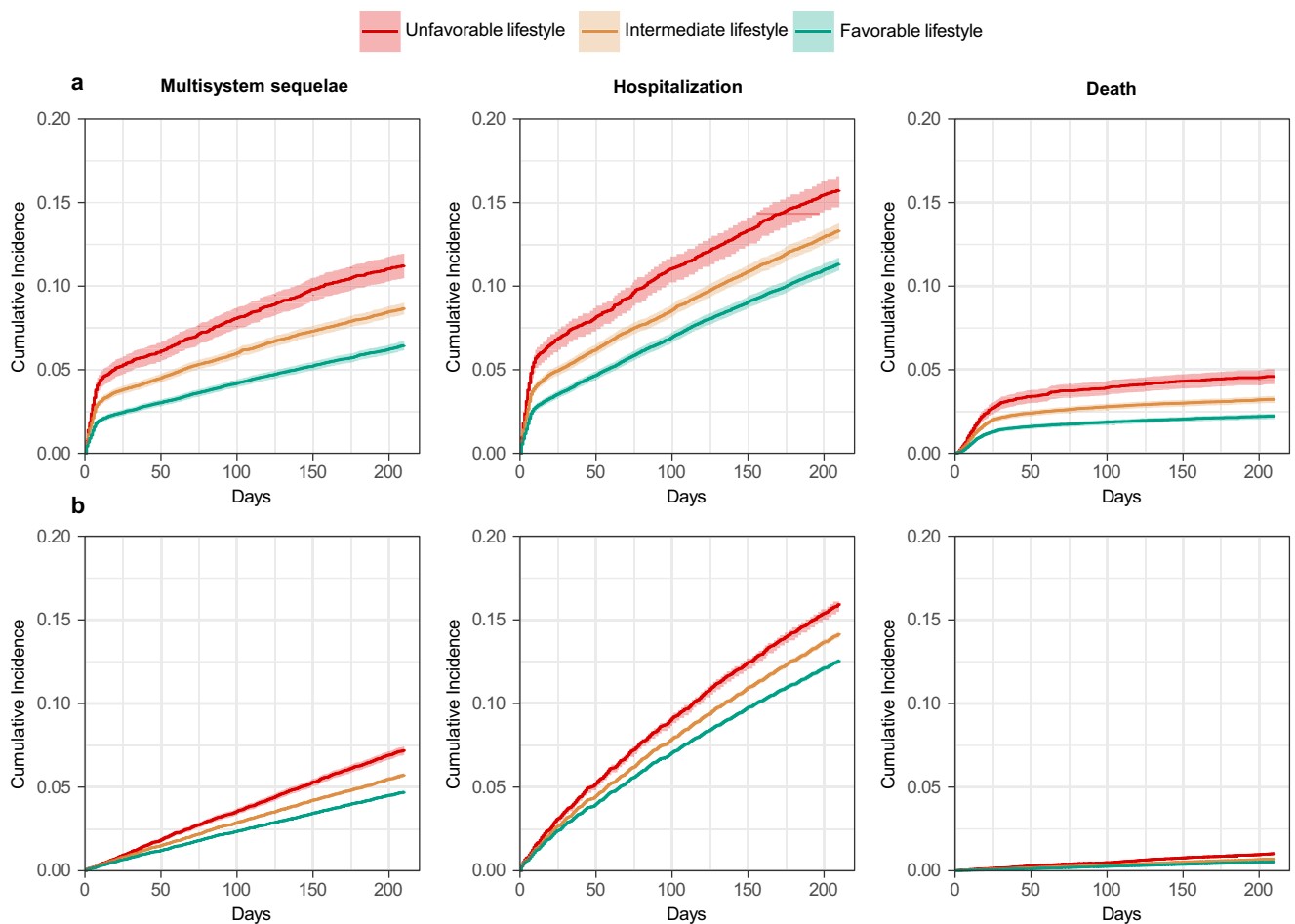

**Fig. 5 | Cumulative incidence curves of composite multisystem sequelae, death, and hospitalization among participants with and without SARS-CoV-2 infection. a** Participants with SARS-CoV-2 infection. **b** Participants with no evidence of SARS-CoV-2 infection. Outcomes were ascertained 0–210 days after SARS-CoV-2 infection. Event rates are presented for the unfavorable lifestyle category (red), the intermediate lifestyle category (orange), and the favorable lifestyle category (green). The shadow of cumulative incidence curves represents 95% CIs.

classification of lifestyle factors are provided in Supplementary Table 1. Briefly, healthy lifestyle components including past or never smoker, moderate alcohol intake ($\leq 4$ times week), BMI < 30 kg/m², at least 150 min of moderate or 75 min of vigorous physical activity per week, less sedentary time (< 4 h per day), healthy sleep duration (7–9 h per day), adequate intake of fruit and vegetables ($\geq 400$ g/day), adequate oily fish intake ($\geq 1$ portion/week), moderate intake of red meat ($\leq 4$ portion week) and processed meat ($\leq 4$ portion week) were defined, in accordance with previous evidence or UK national health service guidelines[55,56].

A binary variable was created for each of the 10 factors, with 1 point assigned for those meeting the healthy criteria and 0 otherwise. A composite lifestyle score was then calculated for each participant by summing the total number of healthy lifestyle factors, ranging from 0 to 10. Based on the composite score, participants were classified into three lifestyle categories: unfavorable (0–5), intermediate (6–7), and favorable (8–10). The lifestyle score was also used as a continuous variable of number of healthy lifestyle factors. Similar methods of defining lifestyle score have been used in the same UK Biobank cohort[57] as well as external cohorts[16,28]. Distributions of lifestyle score and categories are provided in Supplementary Table 2.

The median [IQR] duration between baseline assessment of lifestyle factors and the date of infection was 12.5 [11.8–13.3] years. Part of participants took part in up to two further touchscreen interviews with lifestyle and health-related factors similarly measured. There were generally stable responses to lifestyle factors between baseline

assessment and the latest repeat assessment (median time difference from baseline, 8 years) as shown in Supplementary Fig. 2. 34.9% of participants with an unfavorable lifestyle, 48.6% with an intermediate lifestyle, and 73.7% with a favorable lifestyle at baseline remained in the same corresponding lifestyle category at the latest repeat assessment following a median of 8 years. Overall, the proportion of stable lifestyle categories is 60.6%.

### Outcomes
The outcomes after COVID-19 were prespecified, including a set of multisystem sequelae, death, and hospital admission following the SARS-CoV-2 infection. The multisystem sequelae were selected and defined based on previous evidence of the long COVID, including 75 systemic diseases or symptoms in 10 organ systems: cardiovascular[46], coagulation and hematologic[46], metabolic and endocrine[44], gastrointestinal[48], kidney[43], mental health[45], musculoskeletal[47], neurologic[47], and respiratory disorders[10,13,14], and general symptoms of fatigue and malaise[3,4,42,49]. Detailed definitions of multisystem sequelae are listed in the Supplementary Table 3. Outcomes were identified as follows: individual sequela from the hospital inpatient ICD-10 (*International Classification of Diseases 10th Revision*) diagnosis codes, deaths from the records of national death registry, and hospital admission from hospital inpatient data from the Hospital Episode Statistics. Incident outcomes were assessed in participants with no history of the related outcome within one year before the date of the first infection.

As SARS-CoV-2 infection has been associated with both multi-system manifestations during its acute phase and with sequelae during its post-acute phase[7,49], we conducted analyses stratified by phase of infection. We reported risk of each outcome during the acute phase ($T_O$ to $T_O + 30d$), post-acute phase ($T_O + 30d$ to $T_O + 210d$), and overall period following infection ($T_O$ to $T_O + 210d$) to reflect the full spectrum of post-COVID conditions. The end of follow-up for the overall cohort was September 30, 2022, with the maximum follow-up period censored to 210 days.

## Covariates

We prespecified a list of covariates for adjustment or stratification based on literature review and prior knowledge: socio-demographic characteristics including age, sex, education level (mapped to the international standard for classification of education), index of multiple deprivation (IMD, a summary measure of crime, education, employment, health, housing, income, and living environment)[58], and race and ethnicity; and infection related factors including healthcare settings of the testing (community/outpatient vs inpatient setting as proxy of severity of infection), COVID-19 vaccination status, and SARS-CoV-2 variants.

## Statistical analysis

Baseline characteristics of the overall cohort of participants with SARS-CoV-2 infection and by composite healthy lifestyle categories were reported as mean and standard deviation or frequency and percentage, when appropriate. Multivariable cox proportional hazard (PH) model was used to assess the association between composite healthy lifestyle and risk of multisystem sequelae (composite or by organ systems), death, and hospital admission, with adjustment for age, sex, ethnicity, education level, and IMD. PH assumption across lifestyle categories was tested by Schoenfeld residuals with no violations observed for outcomes. Hazard ratio (HR) and absolute risk reduction (ARR, difference in incidence rate between lifestyle groups per 100 persons during the corresponding follow-up period) were estimated from the Cox model. We also assessed the association between individual lifestyle factor instead of composite categories (each component as a categorical variable with or without mutual adjustment for others, or the number of factors as continuous variables) and risk of outcomes.

We conducted causal mediation analysis[59,60] to quantify the extent to which the habitual healthy lifestyle may affect COVID-19 sequelae through the potential pathway of relevant pre-infection medical conditions (mediator), with the proportion of direct and indirect effects estimated by quasi-Bayesian Monte Carlo methods with 1000 simulations for each. Detailed modeling procedures and a directed acyclic graph are provided in Supplementary Methods.

We examined the association between composite healthy lifestyle and the overall risk of multisystem sequelae in prespecified clinical subgroups by demographic and infection-related factors. The demographic factors included age ($\leq 65$ and $>65$ years), sex (male and female), and ethnicity (White and other ethnic groups). As the risk profile of COVID sequelae was related to vaccination and severity of infection, and may change with the evolving pandemic, infection-related factors including vaccine status (no or one-dose partial vaccination and two-dose full vaccination), test setting (inpatient and outpatient or community), dominant variants during the study period (wildtype, Alpha, Delta, and Omicron BA.1) were assessed. Multiplicative interactions between the composite healthy lifestyle and the stratification variables were tested, with $P$-value reported.

We conducted multiple sensitivity analyses to assess the robustness of primary findings. First, to reflect the multisystem and potentially comorbid nature of COVID sequelae, accounting for both the number of sequelae by an individual and the relative health impact of each sequela. Weights based on Global Burden of Disease study data and methodologies for general diseases and long COVID were assigned to each sequela (Supplementary Table 1)[61,62]. The weighted score was calculated for each participant by summing the weights of all incident sequelae during the follow-up period. Zero inflated Poisson regression was then used to calculate relative risk (RR), with follow-up time set as the offset of the model and adjustment for covariates. Second, to further account for potential reverse causality and more accurately define incident cases, extending the washout period for outcomes from one year to two years. Third, defining events of post-acute sequelae 90 days after infection (follow-up period $T_O + 90d$ to $T_O + 210d$), instead of 30 days in the main analyses. The adjustment was made as there is no uniform definition for long COVID, which is currently described as conditions occurring 30–90 days after infection in existing guidelines[27]. Fourth, restricting the identification of outcomes to the first three ICD diagnoses, which are the main causes for each hospital admission. Fifth, reconstructing a composite lifestyle index without BMI and assessed its association with outcomes. Finally, we conducted quantitative sensitivity analysis to adjust for changes in lifestyle factors over time since the baseline assessment. We used odds ratios to quantify associations and assumed a sensitivity and specificity of 90% for each lifestyle component (Supplementary Methods).

As a healthy lifestyle is associated with a lower risk of chronic diseases and mortality among the general population predated pandemic, we conduct exploratory analysis to compare the effects of healthy lifestyle on adverse outcomes following COVID-19 with the effects among participants without infection. A random index date was assigned to the participants without infection based on the distribution of $T_O$ among those with infection, and we repeated the main analyses with the maximum follow-up period censored to 210 days.

Statistical significance was determined by a 95% confidence interval (CI) that excluded 1 for ratios and 0 for rate differences. All analyses and data visualizations were conducted using R statistical software (version 4.2.2).

## Reporting summary

Further information on research design is available in the Nature Portfolio Reporting Summary linked to this article.

## Data availability

The data that support the findings of this study are available from the UK Biobank. Researchers can apply to use the UK Biobank dataset by registering and applying at https://ukbiobank.ac.uk/register-apply/. The aggregated data supporting the findings of this study are available within the paper and its supplementary information files.

## Code availability

Analysis code used for this study is available at https://github.com/xjq8065524/Lifestyle_Long_COVD_prevention.git.

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

## Acknowledgements

This study was based on data from UK Biobank. Dr Wang is funded through the Clarendon Fund Scholarship. Dr Xie is funded through Jardine-Oxford Graduate Scholarship and a titular Clarendon Fund Scholarship. The research was partially supported by the Oxford National Institute for Health and Care Research (NIHR) Biomedical Research Centre. Dr Prieto-Alhambra is funded through an NIHR Senior Research Fellowship (grant SRF-2018-11-ST2-004). We thank Dr Hao Chen for reviewing the revised manuscript. The funding organizations had no role in the design and conduct of the study; collection, management, analysis, and interpretation of the data; preparation, review, or approval of the manuscript; and decision to submit the manuscript for publication. The views expressed in this publication are those of the author(s) and not necessarily those of the NHS, the NIHR or the Department of Health.

## Author contributions

Y.W. and J.X. had full access to all the data in the study and take responsibility for the integrity of the data and the accuracy of the data analyses. Y.W., J.X. and D.P.-A. conceptualized and designed the study, and acquired, analyzed or interpreted data. Y.W. drafted the manuscript. B.S, M.A.-H., N.L.-B, Y.T, C.L., N.J.-W., and R.P., participated in the discussions and interpretation of the results. All authors critically revised the manuscript for important intellectual content. J.X. conducted statistical analysis. D.P.-A. obtained funding, provided administrative, technical or material support, and supervised the project.

## Competing interests

D.P.-A. reported grants from Amgen, UCB Biopharma, Les Laboratoires Servier, Novartis, and Chiesi-Taylor, as well as speaker fees and advisory board membership with AstraZeneca and Johnson and Johnson outside the submitted work, in addition to research support from Janssen. R.P. has participated in advisory boards for Gilead, MSD, ViiV Healthcare, Theratechnologies and Lilly. His institution has received research support from Gilead, MSD, and ViiV Healthcare. The remaining authors declare no competing interests.
