## [Peer Review File · Nature Communications]

Modifiable lifestyle factors and the risk of post-COVID-19 multisystem sequelae, hospitalization, and deathREVIEWER COMMENTS

Reviewer #1 (Remarks to the Author):

This is a very well executed idea to investigate the role of pre-existing lifestyle attitudes and risk of post-covid 19 condition, using the UK Biobank.

I have the following comments:

1. How were patients that died during/ following acute infection considered, as dead people cannot get long covid.

2. Regarding vaccination status: The HR of fully vaccinated subjects is significantly lower than the HR of the no or partially vaccinated subjects for all the outcomes.

Would be necessary to examine the risk of outcomes by vaccination status having 3 categories (not vaccinated, partially vaccinated, fully vaccinated) and reported accordingly. In addition, an individual assessment of lifestyle in the fully vaccinated individuals in the online material would be helpful.

3. Do these lifestyle factors have any incremental association compared to use of BMI alone?

4. "The reduced risk of outcomes was observed in participants who received two doses of vaccine (breakthrough infection) and those who were unvaccinated or partially 1-dose vaccinated (non breakthrough infection)". I think "and those" could change to "compared to those" for increased clarity- but again divide in 3 categories re vaccination

5. It is mentioned that the healthy lifestyle was associated with lower risk of post COVID-19 condition, independently of co-morbidities. Can you say that with certainty for all the co-morbidities examined? The indirect effect of lifestyle in some co-morbidities (i.e. diabetes, cardiovascular disease) is much bigger compared to that of others (i.e. coagulation). Do you think the differences in the effect of the co-morbidities examined influenced the overall result?

6. It would be useful to know the number of participants with each of the co-morbidities examined so as to know the size of each group (can be given in a supplementary table).

7. In Figure 2, 'General' is listed as a pre-infection condition. Which system/what diseases does this include?

8. In the discussion, it may be helpful to discuss the role that chronic inflammation and immunity may have in the post COVID-19 condition and how this may be affected by nutrition and healthy lifestyle. There are a few papers on the potential role of nutrition that could be referenced here (PMID: 34308134, PMID: 38381595. Chronic inflammation, nutrition and chronic diseases are inexplicably interlinked and may explain – to some extent – the findings of this study.

9. The majority of the study subjects were in the intermediate and favourable lifestyle categories. This is an important limitation and unavoidable selection bias that may have a significant impact on the results and should be mentioned in the limitations.

10. The authors acknowledge very subtly in the limitations that the assessment of the lifestyle patterns predated quite significantly the time of the infection. Can we please see an average/ median time from the time of the latest questionnaire confirmation and the infection to be able to assess how big a limitation this is? Also, the authors say that lifestyle patterns do not really change from initial assessment and subsequent assessments. Can you please quantify this?

11. Regarding alcohol and smoking- can the authors provide an explanation why such results are seen?

Reviewer #1 (Remarks on code availability):

I do not think the authors have shared their statistical coding for this paper

Reviewer #2 (Remarks to the Author):

Reviewer #3 (Remarks to the Author):

Thank you so much for inviting me to review the paper "Healthy lifestyle for the prevention of post-COVID-19 multisystem sequelae, hospitalization, and death: a prospective cohort study". The paper has many strengths, among them the thorough literature review and appropriate methods that are enhanced with robustness checks via sensitivity analysis. There are some minor suggestions that I have (some of them I mention below), but I believe there is a major issue that needs to be addressed.

To me, the most interesting and relevant to population health findings are those presented in eTable5 and Figure 4. It's the comparison of the association between healthy lifestyle and the various outcomes stratified by C19 status that matters. In other words, the question of population health interest is whether healthy lifestyle has additional (or less) benefits in those with C19 compared to those without. Presenting the main analysis only in those with C19 confirms that healthy lifestyle is beneficial (protective) but while this is interesting, it's the comparison with the "control group" (those without C19) that matters. To this end, although I am not against the parallel analysis, I think a straightforward C19 by healthy lifestyle interaction in the pooled sample (C19 plus no C19) would be a good idea. My recommendation is that the pooled sample analysis becomes the main analysis that is presented and the C19 only results are moved to the Appendix. A bit more discussion on why the associations between healthy lifestyle and the various outcomes differ (or not) by C19 status would be needed.

Minor points:

Missing data: Participants with missing data on the study 145 exposures and covariates of interest at baseline were further excluded. How many are those? The consensus in the missing data literature is that if >5% deletion may introduce bias. If so, a principled method (multiple imputation, inverse probability weighting, FIML) to handle missing data would be needed.

Biobank is obviously a selected sample and although the authors mention this in the limitation section, they also claim that despite this the relative risk is not influenced. Why? What is the transportability assumption underlying this claim? Obviously selection bias can lead to relative risks being biased, I believe the claim that they are not should be justified and the strong assumptions underlying this clarified or removed.

Abstract typo: "Adherence to a healthy lifestyle predated pandemic" doesn't sound quite right to me, maybe something like "adherence to a healthy lifestyle that predated the pandemic"?

Reviewer #4 (Remarks to the Author):

This manuscript provides a comprehensive analysis of the relationship between pre-pandemic healthy lifestyle factors and post-COVID-19 sequelae using data from the UK Biobank cohort. Findings from this study demonstrate the clinical and public health importance of adhering to a healthy lifestyle to reduce the overall post-COVID-19 burden and improve preparedness for future pandemics. The authors have performed extensive and complex analytical work, but there are still some methodological issues that need to be resolved before they can be considered for publication.

Introduction: Although the authors provide a detailed description of the study background and research questions, providing specific numbers to support statements and research questions would enhance reader confidence.

Method - Data Sources and Study Cohorts: Excluding participants with missing covariate data from the analysis may introduce selection bias. It's recommended that the authors include these participants and employ appropriate methods to handle missing data (e.g., multiple imputation).

Method - Lifestyle Factors: While obesity is included as a lifestyle factor in the manuscript, the Lancet article referenced by the authors does not classify it as such. While obesity is widely considered to have profound effects on long-term health outcomes, it's preferable to view it as a consequence of unhealthy lifestyle rather than a lifestyle factor itself.

Method - Lifestyle Factors: The authors mention that "Part of participants took part in up to two further touchscreen interviews with lifestyle and health-related factors similarly measured." It's necessary for the authors to specify the number of participants who participated in the first, second, and third assessments and the proportion of participants who underwent 2 or 3 lifestyle assessments. Additionally, numerical representation in the Supplementary Figures supporting this assertion would strengthen the argument, such as indicating the proportion of participants who transitioned from healthy to unhealthy lifestyles.

Method - Outcomes: Linking the UK Biobank to the Hospital Episode Statistics (HES) database to capture health outcomes defined by ICD-10 codes requires clarification on the proportion of participants successfully linked to the HES database. Any potential issues with linkage failure should be discussed in the discussion section.

Method - Outcomes: The choice of a 210-day follow-up endpoint seems inconsistent with the statement: "Increased risk and burden of cardiovascular, pulmonary, neuropsychiatric, and metabolic disorders were reported during the 6 to 12 months following SARS-CoV-2 infection, with persistent risk observed for several diseases up to 2 years." The reason behind this choice needs to be explained given that the research period of this study is from March 1, 2020 to March 1, 2022.

Method - Outcomes: The authors need to provide a more detailed description of censoring. For example, how did they handle participants who died or withdrew from the UK Biobank?

Method - Covariates: In addition to the covariates listed by the authors, other important confounding factors such as comorbidities, biomarkers, medications, and index month should be considered as well.

Method - Statistical Analysis: The term "Hazard ratio (HR) and absolute risk reduction (ARR, difference in incidence rate between lifestyle groups per 100 persons during the corresponding follow-up period)" might be intended to refer to "person-month" or "person-day." Normally, reporting the number of events, person-month/person-day, and incidence rates of each group as well as crude HRs is required.

Method - Statistical Analysis: Death is an important competing event in this study. It's recommended that the authors consider a Competing Risk Cox proportional hazard regression model or incorporate inverse-probability-of-censoring weighting into the Cox model.

Method - Statistical Analysis: A crucial assumption for mediation analysis is that the mediator must occur after exposure. Please provide more information on the technical details about this.

Method - Statistical Analysis: Regarding extending the washout period for outcomes from one year to two years, please clarify the definition of the washout period. It's assumed the authors refer to "one year before the date of the first infection." However, this washout period may not be meaningful for incurable chronic conditions. Using a "latency period" (one year after exposure assessment date) may be more helpful for accounting for potential reverse causality.

Method - Statistical Analysis: Please explain what is meant by "the first three ICD diagnoses." There may be a discrepancy in understanding of the first X number of ICD diagnoses in HES between the authors and the reviewer.

Method - Statistical Analysis: Exploratory analysis of non-infected individuals may not be appropriate. Firstly, it's irrelevant to the study question. Secondly, while these individuals have no infection records, it's likely they were not assessed/recorded rather than being uninfected.

Results: Please provide a flowchart.

Reviewer #4 (Remarks on code availability):

I did not recognise that the code was available.

REVIEWER COMMENTS

Reviewer #1 (Remarks to the Author):

This is a very well executed idea to investigate the role of pre-existing lifestyle attitudes and risk of post-covid 19 condition, using the UK Biobank.

Response: We thank the reviewer for the thorough assessment of our manuscript and recognition of the importance of this study.

I have the following comments:

1. How were patients that died during/ following acute infection considered, as dead people cannot get long covid.

Response: We included participants who were alive by March 1, 2020 (date of the first recorded case in the UK Biobank) and had a positive SARS-CoV-2 PCR test between March 1, 2020 and March 1, 2022. Death is considered in our analyses mainly in three ways:

- For the outcome of multisystem sequelae (defined as incident diseases identified by ICD10 codes), we used cause-specific Cox proportional hazards regression models where death was considered as a competing risk (censored if dead) to estimate the risk.
- We also assess the risk all-cause mortality as a key composite outcome post-infection.
- We divided the post-infection period into acute (T_0 to $T_0 + 30d$) and post-acute phases ($T_0 + 30d$ to $T_0 + 210d$). This means participants were excluded if they died during the acute period when assessing the post-acute risk.

To clarify, we add sentences in the Method: “*Death was considered as a competing risk in the analyses of sequelae outcomes.*”

2. Regarding vaccination status: The HR of fully vaccinated subjects is significantly lower than the HR of the no or partially vaccinated subjects for all the outcomes.

Would be necessary to examine the risk of outcomes by vaccination status having 3 categories (not vaccinated, partially vaccinated, fully vaccinated) and reported accordingly. In addition, an individual assessment of lifestyle in the fully vaccinated individuals in the online material would be helpful.

Response: We thank the reviewer for these suggestions. We agree with the reviewer that a more detailed category would be more helpful. However, limited number of participants ($n=1.66\%$) were in the “partially vaccination” group prevent from analyzing it separately. We had provided the risk estimates of lifestyle in fully vaccinated individuals in subgroup analyses (**Table 2**, please see analyses by Vaccine status). The associations between healthy lifestyle and lower risk of post-COVID conditions (PCC) were consistently observed across subgroups by vaccination, and no significant interaction effect was detected.

3. Do these lifestyle factors have any incremental association compared to use of BMI alone?

Response: We thank the reviewer for these suggestions. We conducted additional analyses by first re-constructing a composite lifestyle index without BMI and then assessed its association with outcomes. The results are consistent albeit appearing to be less pronounced.

Composite outcomes	HR (95% CI)	
	Intermediate	Favorable
Multisystem sequelae	0.84 (0.76 to 0.93)	0.77 (0.70 to 0.85)
Death	0.77 (0.67 to 0.89)	0.64 (0.56 to 0.74)
Hospitalization	0.93 (0.85 to 1.02)	0.90 (0.82 to 0.98)

4. “The reduced risk of outcomes was observed in participants who received two doses of vaccine (breakthrough infection) and those who were unvaccinated or partially 1-dose vaccinated (non breakthrough infection)”. I think “and those” could change to “compared to those” for increased clarity- but again divide in 3 categories re vaccination

Response: We thank the reviewer for these suggestions. This sentence is to indicate that the adverse association between healthy lifestyle and PCC risk were observed in both fully vaccinated and not or partially vaccinated participants, suggesting the associations observed were independent of vaccination status. We are not intended to compare the PCC risk across vaccination and the comparison groups are different lifestyle category (Favorable/Intermediate lifestyle compared to unfavorable one in those with/without vaccination).

5. It is mentioned that the healthy lifestyle was associated with lower risk of post COVID-19 condition, independently of co-morbidities. Can you say that with certainty for all the co-morbidities examined? The indirect effect of lifestyle in some co-morbidities (i.e. diabetes, cardiovascular disease) is much bigger compared to that of others (i.e. coagulation). Do you think the differences in the effect of the co-morbidities examined influenced the overall result?

Response: We thank the reviewer for these suggestions.

We used same definition for pre-infection chronic diseases as post-infection sequelae consisting of 75 diseases/symptoms in 10 organ systems to estimate the direct effect of healthy lifestyle independent of the pathway of healthy lifestyle -> pre-infection comorbidities -> PCC. The proportion of direct protective effect of healthy lifestyle for diabetes is 56% vs 71% for any sequelae. This heterogeneous mediation proportions across disease in organ systems are expected considering the multisystem nature of PCC, and potential underlying mechanisms linking healthy lifestyle, infection, and PCC. Overall, we observed the majority of protective effect of healthy lifestyle are independent of pre-infection comorbidity (from 56% for diabetes to 93% for fatigue; 71% for any sequelae).

We have modified the statement to improve its scientific accuracy: “*Healthy lifestyle was associated with lower risk of post COVID-19 condition, independently of the corresponding co-morbidity prior the infection.*”

One of the novelties in our study design was the ability to dissect the protection of a healthy lifestyle. It is biologically plausible to see varying proportions of indirect effects from lifestyle depending on the specific outcome of interest. “*For example, pre-existing diabetes had the strongest association with post-infection diabetes sequela, therefore lifestyle is more likely to confer its protection indirectly through this pathway. In contrast, thrombotic events are generally more acute and transient, making it less likely for lifestyle factors to confer protection through an indirect mechanism.*” We have interpreted these points further in the discussion.

6. It would be useful to know the number of participants with each of the co-morbidities examined so as to know the size of each group (can be given in a supplementary table).

Response: We thank the reviewer for these suggestions. Because it is overwhelmed to put all these numbers into the figure, and we decided to report them in the supplementary table.

Pre-infection medical conditions	Participants (%)
Any complication	42.86
General fatigue	1.29
Coagulation diseases	5.01
Neurologic diseases	11.57
Pulmonary diseases	3.56
Kidney diseases	5.64
Gastrointestinal diseases	17.51
Mental health diseases	12.25
Musculoskeletal diseases	14.03
Cardiovascular diseases	10.44
Diabetes	7.54

7. In Figure 2, 'General' is listed as a pre-infection condition. Which system/what diseases does this include?

Response: We thank the reviewer for these suggestions. We now clarify it as "General fatigue".

8. In the discussion, it may be helpful to discuss the role that chronic inflammation and immunity may have in the post COVID-19 condition and how this may be affected by nutrition and healthy lifestyle. There are a few papers on the potential role of nutrition that could be referenced here (PMID: 34308134, PMID: 38381595). Chronic inflammation, nutrition and chronic diseases are inexplicably interlinked and may explain – to some extent – the findings of this study.

Response: We thank the reviewer for these suggestions. We add additional sentences on the potentially involved mechanisms of healthy lifestyle in the prevention of PCC from the perspectives of chronic inflammation and immunity in discussion: *"Biologically, the overlapping mechanisms between unhealthy lifestyle and viral infection and post-infection conditions may also be involved. Favorable lifestyle factors, such as physical activity and healthy diet, confer health benefits including protection against inflammation,^{1,2} autoimmunity,^{3,4} and clotting abnormality,⁵ which are implicated in the potential pathogenesis of long COVID.^{6"}*

9. The majority of the study subjects were in the intermediate and favourable lifestyle categories. This is an important limitation and unavoidable selection bias that may have a significant impact on the results and should be mentioned in the limitations.

Response: We thank the reviewer for these suggestions. We add extra discussion for the potential bias it may cause in the limitation and potential explanation: *"Second, the majority of participants (>85%) were classified as intermediate or favorable lifestyle category, suggesting that the study population appears to be healthier than the wider general population. However, the exact distribution of lifestyle categories based on the similar 10 modifiable factors in another population remains unknown. A previous study in Australia (n=231,048)⁷ reported 68% of participants had ≤ 1 unhealthy lifestyle factor out of 7 factors assessed (smoking, alcohol consumption, physical activity, sitting time, sleeping duration and diet), which partly in line with our results. Assuming participants are healthier than the UK general population, the absolute risk estimates such as ARR should be interpreted with caution. Nevertheless, the relative associations of risk factors with disease outcomes in UK Biobank were tested to be generalizable and comparable to those from other representative cohort of general population.^{8,9} In addition, this high proportion could also suggest potential misreporting bias in self-reported lifestyle data. However, healthy reporting bias may be more common in socioeconomically deprived individuals¹⁰ and tends to bias any genuine association towards null. Self-reported lifestyle data, such as sleep duration¹¹ and physical activity,¹² have been shown to be highly correlated with accelerometer-derived measures in UK Biobank."*

10. The authors acknowledge very subtly in the limitations that the assessment of the lifestyle patterns predated quite significantly the time of the infection. Can we please see an average/ median time from the time of the latest questionnaire confirmation and the infection to be able to assess how big a limitation this is? Also, the authors say that lifestyle patterns do not really change from initial assessment and subsequent assessments. Can you please quantify this?

Response: We thank the reviewer for these suggestions. We provided additional information that median duration between baseline assessment of lifestyle factors and the date of infection (median [IQR]: 12.5 [11.8-13.3] years). We also quantify the change in lifestyle factors across repetitive visits in the main text and supplementary: "34.9% of participants with an unfavorable lifestyle, 48.6% with an intermediate lifestyle, and 73.7% with a favorable lifestyle at baseline remained in the same corresponding lifestyle category at the latest repeat assessment following a median of 8 years. Overall, the proportion of stable lifestyle categories is 60.6%."

11. Regarding alcohol and smoking- can the authors provide an explanation why such results are seen?

Response: We thank the reviewer for these suggestions. The associations of individual lifestyle factor such as smoking with health outcomes are in line with previous evidence suggesting that smoking is a strong risk factor for both infection and PCC in hospitalized and nonhospitalized patients.¹³ By contrast, previous evidence did not consistently identify alcohol consumption as a risk factor for PCC^{13,14} and its association with chronic diseases was controversial in previous evidence. For many chronic diseases, including those considered to be alcohol-related by WHO (e.g., CHD and diabetes), uncertainty remains about their association with drinking. We add in the limitations that: “..given the potential non-linear effects of lifestyle factors, such as alcohol consumption, caution is warranted when interpreting associations between binarized lifestyle factors and outcomes. *The health effects and recommended targets of several individual lifestyle factors, such as alcohol consumption and red meat intake, are inconsistent in previous epidemiological studies and guidelines,^{15,16} and may potentially vary by disease outcome of interest. Findings on the association of such individual factors with post-infection complications should be considered in the wider context of chronic disease, whether directly related to infection or not.*”

Reviewer #1 (Remarks on code availability):

I do not think the authors have shared their statistical coding for this paper

Response: We thank the reviewer for these suggestions. We add the **Code Availability** part and the Analysis code used for this study will be deposited in a GitHub repository at the time of publication.

Reviewer #2 (Remarks to the Author):

Response: We thank the reviewer for the thorough assessment of our manuscript.

Reviewer #3 (Remarks to the Author):

Thank you so much for inviting me to review the paper “Healthy lifestyle for the prevention of post-COVID-19 multisystem sequelae, hospitalization, and death: a prospective cohort study”. The paper has many strengths, among them the thorough literature review and appropriate methods that are enhanced with robustness checks via sensitivity analysis. There are some minor suggestions that I have (some of them I mention below), but I believe there is a major issue that needs to be addressed.

To me, the most interesting and relevant to population health findings are those presented in eTable5 and Figure 4. It's the comparison of the association between healthy lifestyle and the various outcomes stratified by C19 status that matters. In other words, the question of population health interest is whether healthy lifestyle has additional (or less) benefits in those with C19 compared to those without. Presenting the main analysis only in those with C19 confirms that healthy lifestyle is beneficial (protective) but while this is interesting, it's the comparison with the “control group” (those without C19) that matters. To this end, although I am not against the parallel analysis, I think a straightforward C19 by healthy lifestyle interaction in the pooled sample (C19 plus no C19) would be a good idea. My recommendation is that the pooled sample analysis becomes the main analysis that is presented and the C19 only results are moved to the Appendix. A bit more discussion on why the associations between healthy lifestyle and the various outcomes differ (or not) by C19 status would be needed.

Response: We thank the reviewer for these suggestions. The main aim of the current research is to investigate whether adherence to a healthy lifestyle that predated the pandemic is associated with decreased risk of multisystem complications, death and hospitalization following the SARS-CoV-2 infection, not to test whether the health effects of healthy lifestyle differed by COVID-19 status.

Although the risk of diseases or symptoms were increased following COVID-19, referred to as long COVID or PCC, its pathogenesis remains uncertain and currently there is no treatment available for long COVID that tested by clinical trial. This means that despite a disease with the same term, such as depression in the uninfected population and depression following COVID-19 as a sequela, may have different underlying aetiologies, especially considering that the latter may be due to the infection itself.

However, we agreed with reviewer that understanding the how the estimates may differ could be of clinical interest and therefore designed a parallel comparison.

It should be noted that the same disease between infected vs uninfected groups may have different aetiologies and we observed that lifestyle has stronger effects on multisystem sequelae in the infected group during the acute phases (0-30d following infection), suggesting the particular benefits of healthy lifestyle in preventing acute outcomes that could be directly and biologically caused by viral infection.

Minor points:

Missing data: Participants with missing data on the study exposures and covariates of interest at baseline were further excluded. How many are those? The consensus in the missing data literature is that if >5% deletion may introduce bias. If so, a principled method (multiple imputation, inverse probability weighting, FIML) to handle missing data would be needed.

Response: We thank the reviewer for these suggestions. We excluded participants with missing data on exposure and

covariates of age and sex with low percentage of missing while we did handle other covariates with higher missingness. The manuscript was drafted by first author and statistical analyses were mainly run by last author. We are sorry for the omission and have further clarify the specific methods used for handling missing data:

“We addressed missing data using the following approaches: (1) Participants with missing values in age and sex (<0.1%) were excluded. (2) Participants with missing values in Ethnicity were classified as “Other ethnic groups”. (3) Participants with missing values in Education level (0.9%) were classified as “Category I”, which includes “None of the above” and “Prefer not to answer”. (4) Missing values in IMD (13.8%) were imputed with a mean value of the entire UK Biobank cohort.” We have added these in the Methods.

Information on the missing covariates:

Covariates	Percentage with missing value
Age	<0.1%
Sex	<0.1%
Ethnicity	14.0%
Education level	0.9%
IMD	13.8%

Biobank is obviously a selected sample and although the authors mention this in the limitation section, they also claim that despite this the relative risk is not influenced. Why? What is the transportability assumption underlying this claim? Obviously selection bias can lead to relative risks being biased, I believe the claim that they are not should be justified and the strong assumptions underlying this clarified or removed.

Response: We thank the reviewer for these suggestions. The relative risk of disease outcome with exposure derived from UKB has been tested to be generalizable to other representative, general population based cohort (Batty GD, et al. Comparison of risk factor associations in UK Biobank against representative, general population based studies with conventional response rates: prospective cohort study and individual participant meta-analysis. BMJ.). While the absolute risk should be interpreted with cautious if the study participants were healthier than wider general population. We add further discussion for this in the limitations section: *“Assuming participants are healthier than the UK general population, the absolute risk estimates such as ARR should be interpreted with caution. Nevertheless, the relative associations of risk factors with disease outcomes in UK Biobank were tested to be generalizable and comparable to those from other representative cohort of general population.”*⁸

Abstract typo: “Adherence to a healthy lifestyle predated pandemic” doesn’t sound quite right to me, maybe something like “adherence to a healthy lifestyle that predated the pandemic”?

Response: We thank the reviewer for these suggestions. We have changed it to “adherence to a healthy lifestyle that predated the pandemic...”.

Reviewer #4 (Remarks to the Author):

This manuscript provides a comprehensive analysis of the relationship between pre-pandemic healthy lifestyle factors and post-COVID-19 sequelae using data from the UK Biobank cohort. Findings from this study demonstrate the clinical and public health importance of adhering to a healthy lifestyle to reduce the overall post-COVID-19 burden and improve preparedness for future pandemics. The authors have performed extensive and complex analytical work, but there are still some methodological issues that need to be resolved before they can be considered for publication.

Response: We thank the reviewer for the thorough assessment of our manuscript and recognition of the importance of this study.

Introduction: Although the authors provide a detailed description of the study background and research questions, providing specific numbers to support statements and research questions would enhance reader confidence.

Response: We thank the reviewer for these suggestions. We now provided additional specific numbers and risk estimates based on previous evidence to strengthen our statements and research background: *“Previous studies on its prevention has mainly focused on vaccination and pharmaceutical approaches, including antivirals (e.g., molnupiravir and nirmatrelvir) and other drugs repurposed for long COVID (e.g., metformin). Increasing evidence suggests that vaccination before infection and use of antivirals during acute phase in selected high-risk patients only partially mediate the risk of long COVID at 6 to 12 months following infection (by 15%-51% for vaccination, by 26% for nirmatrelvir, and by 14% for molnupiravir). Several potential drugs for long COVID are still under investigation without yielding reliable results. Evidence for the non-pharmaceutical management strategies are also lacking...”*

Method - Data Sources and Study Cohorts: Excluding participants with missing covariate data from the analysis may introduce selection bias. It's recommended that the authors include these participants and employ appropriate methods to handle missing data (e.g., multiple imputation).

Response: We thank the reviewer for these suggestions. We excluded participants with missing data on exposure and covariates of age and sex with low percentage of missing while we did handle other covariates with higher missingness. The manuscript was drafted by first author and statistical analyses were mainly run by last author. We are sorry for the omission and have further clarify the specific methods used for handling missing data:

“We addressed missing data using the following approaches: (1) Participants with missing values in age and sex (<0.1%) were excluded. (2) Participants with missing values in Ethnicity were classified as “Other ethnic groups”. (3) Participants with missing values in Education level (0.9%) were classified as “Category I”, which includes “None of the above” and “Prefer not to answer”. (4) Missing values in IMD (13.8%) were imputed with a mean value of the entire UK Biobank cohort.” We have added these in the Methods.

Information on the missing covariates:

Covariates	Percentage with missing value
Age	<0.1%
Sex	<0.1%
Education level	14.0%
Ethnicity	0.9%
IMD	<0.1%

Method - Lifestyle Factors: While obesity is included as a lifestyle factor in the manuscript, the Lancet article referenced by the authors does not classify it as such. While obesity is widely considered to have profound effects on long-term health outcomes, it's preferable to view it as a consequence of unhealthy lifestyle rather than a lifestyle factor itself.

Response: We thank the reviewer for these suggestions. It would be very challenging to justify whether BMI should

treated be an element of lifestyle or an outcome of lifestyle. In literature, both definitions has been frequently used: (add relevant paper here).

As suggested by the reviewer, we conducted analyses by first re-constructing a composite lifestyle index without BMI and then assessed its association with outcomes. The results consistently suggested that adherence to a healthy lifestyle was associated with substantially lower risk of complications across organ systems, death, and hospitalization following COVID-19.

Composite outcomes	HR (95% CI)	
	Intermediate	Favorable
Multisystem sequelae	0.84 (0.76 to 0.93)	0.77 (0.70 to 0.85)
Death	0.77 (0.67 to 0.89)	0.64 (0.56 to 0.74)
Hospitalization	0.93 (0.85 to 1.02)	0.90 (0.82 to 0.98)

Method - Lifestyle Factors: The authors mention that "Part of participants took part in up to two further touchscreen interviews with lifestyle and health-related factors similarly measured." It's necessary for the authors to specify the number of participants who participated in the first, second, and third assessments and the proportion of participants who underwent 2 or 3 lifestyle assessments. Additionally, numerical representation in the Supplementary Figures supporting this assertion would strengthen the argument, such as indicating the proportion of participants who transitioned from healthy to unhealthy lifestyles.

Response: We thank the reviewer for these suggestions. The participants with available data on the first repetitive survey is only about 20,000 and we used the latest one including more than 60,000 participants. We have quantified the change in lifestyle factors across repetitive visits in the main text and supplementary Figure 2: "34.9% of participants with an unfavorable lifestyle, 48.6% with an intermediate lifestyle, and 73.7% with a favorable lifestyle at baseline remained in the same corresponding lifestyle category at the latest repeat assessment following a median of 8 years. Overall, the proportion of stable lifestyle categories is 60.6%."

Method - Outcomes: Linking the UK Biobank to the Hospital Episode Statistics (HES) database to capture health outcomes defined by ICD-10 codes requires clarification on the proportion of participants successfully linked to the HES database. Any potential issues with linkage failure should be discussed in the discussion section.

Response: We thank the reviewer for these suggestions. Based on the online document of UK Biobank (<https://biobank.ndph.ox.ac.uk/showcase/refer.cgi?id=138483>) and prior published studies,^{9,17} all participants has been linked to electronic health records including hospital inpatient episodes, cancer registrations, death registrations. The linkage to primary care data was only available for approximately 260,000 participants (45% of the cohort),¹⁷ but we did not use such data in our study.

Method - Outcomes: The choice of a 210-day follow-up endpoint seems inconsistent with the statement: "Increased risk and burden of cardiovascular, pulmonary, neuropsychiatric, and metabolic disorders were reported during the 6 to 12 months following SARS-CoV-2 infection, with persistent risk observed for several diseases up to 2 years." The reason behind this choice needs to be explained given that the research period of this study is from March 1, 2020 to March 1, 2022.

Response: We thank the reviewer for these suggestions. The current data used was censored on Sept 30, 2022 and we use the follow-up to 210-day to guarantee that every participant was follow-up at least 7 months. The current evidence consistently suggests that the risks of diseases across multiple organ systems were increased during 6 to 12 months following COVID-19. However, a few studies reported that such risk extended up to 2 years for several diseases such as dementia and several neurological diseases.

Method - Outcomes: The authors need to provide a more detailed description of censoring. For example, how did they handle participants who died or withdrew from the UK Biobank?

Response: We thank the reviewer for these suggestions. We included participants who were alive by March 1, 2020 (date of the first recorded case in the UK Biobank) and had a positive SARS-CoV-2 PCR test between March 1, 2020 and March 1, 2022, with no withdrew.

Death is considered in our analyses mainly in three ways:

- For the outcome of multisystem sequelae (defined as incident diseases identified by ICD10 codes), we used cause-specific Cox proportional hazards regression models where death was considered as a competing risk (censored if dead) to estimate the risk.
- We also assess the risk all-cause mortality as a key composite outcome post-infection.
- We divided the post-infection period into acute (T_0 to $T_0 + 30d$) and post-acute phases ($T_0 + 30d$ to $T_0 + 210d$). This means participants included in the post-acute phase survived at least 30 days following infection.

To clarify, we add sentences in the Method: "*Death was considered as a competing risk in the analyses of sequelae outcomes.*"

Method - Covariates: In addition to the covariates listed by the authors, other important confounding factors such as comorbidities, biomarkers, medications, and index month should be considered as well.

Response: We thank the reviewer for these suggestions. We accounted for multiple pre-infection comorbidities and observed that the majority of protective effect of healthy lifestyle are independent of pre-infection comorbidity (from 56% for diabetes to 93% for fatigue; 71% for any sequelae). We also accounted for multiple virus-related factors in the subgroups analyses including test setting, variants, and vaccine status. The adjustment for covariable was based on prior evidence, research, and knowledge, and factors mentioned by such as biomarkers and medications were not considered to be of high relevance, and included of variables such as biomarkers may lead to overadjustment, especially considering these markers (as the reviewer did not specific which specific marker, we assume the biomarkers to be blood chemistry and blood count) are highly likely to be in the pathway of inflammation, autoimmunity, and clotting abnormality that identified to mediate the association of lifestyle with health outcomes.

Method - Statistical Analysis: The term "Hazard ratio (HR) and absolute risk reduction (ARR, difference in incidence rate between lifestyle groups per 100 persons during the corresponding follow-up period)" might be intended to refer to "person-month" or "person-day." Normally, reporting the number of events, person-month/person-day, and incidence rates of each group as well as crude HRs is required.

Response: We thank the reviewer for these suggestions. We did reported HRs, number of events/participants, and ARR in the main figure for the association between lifestyle and post-infection outcomes. ARR can be reported as, for example, risk difference between group per 1,000 person-years like mentioned by the reviewer or difference in incidence rate between groups per 100 persons during the corresponding follow-up period. They are actually the same absolute risk measure. However, the latter one used in the manuscript is more clinically relevant. For example, ARR (%) at 210 days are 7.08, 1.99, and 6.14 for the association between favorable healthy lifestyle and sequelae, death, and hospitalization, respectively, which corresponded to an absolute risk reduction of 7.08, 1.99, and 6.14 fewer cases per 100 people at 210 days after infection. It's obviously more evident for reader to better understand the relevance of such absolute risk estimates. Given the extremely broad readership of the Nature Portfolio, including those without specific epidemiological knowledge, we used this form of ARR to better inform readers and policy makers.

Furthermore, this method has also been widely used in previous epidemiological studies:

- Xie Y, Choi T, Al-Aly Z. Molnupiravir and risk of post-acute sequelae of covid-19: cohort study. *BMJ*. 2023;381
- Al-Aly Z, Bowe B, Xie Y. Long COVID after breakthrough SARS-CoV-2 infection. *Nature medicine*. 2022;28(7):1461-1467.

Method - Statistical Analysis: Death is an important competing event in this study. It's recommended that the authors consider a Competing Risk Cox proportional hazard regression model or incorporate inverse-probability-of-censoring weighting into the Cox model.

Response: We thank the reviewer for these suggestions. For the outcome of multisystem sequelae (defined as incident diseases identified by ICD10 codes), we used cause-specific Cox proportional hazards regression models where death was considered as a competing risk (censored if dead) to estimate the risk.

Method - Statistical Analysis: A crucial assumption for mediation analysis is that the mediator must occur after exposure. Please provide more information on the technical details about this.

Response: We thank the reviewer for these suggestions. We have made this clear the mediator (comorbidities) occur after exposure (healthy lifestyle). Please refer to the supplementary methods: *"The mediator of interest was defined as relevant events that occurred between the baseline lifestyle assessment and the date of COVID-19 diagnosis."*

Method - Statistical Analysis: Regarding extending the washout period for outcomes from one year to two years, please clarify the definition of the washout period. It's assumed the authors refer to "one year before the date of the first infection." However, this washout period may not be meaningful for incurable chronic conditions. Using a "latency period" (one year after exposure assessment date) may be more helpful for accounting for potential reverse causality.

Response: We thank the reviewer for these suggestions.

The lifestyle factors measured at baseline (tested stable over time) is the exposure of interest at the infection. In our study, the max follow-up period is 7 months so it's impossible to exclude incident events during the first one year after infection. But we did divide the post-infection into acute (T_0 to $T_0 + 30d$) and post-acute phases ($T_0 + 30d$ to $T_0 + 210d$). The post-acute phases can be accounted as excluding any incident events in the first 30 days after infection. The protective effects of healthy lifestyle were observed during both the acute and post-acute period (that exclude the "latency period").

To account for potential reverse causality, we used one year washout in the main analyses and two year in the sensitivity analyses and observed consistent results. We add definition of washout in the manuscript: "extending the washout period for outcomes from one year to two years (by excluding any outcome of interest two year before the date of the first infection)". In addition, we account all study outcome (for multisystem sequelae, **Fig. 2**) from baseline recruitment (2007-2019) to the time of infection. After accounting for any previous related disease history for multisystem sequelae, more than 70% of protective effects of healthy lifestyle were largely attributable to direct effects of healthy lifestyle, independent of history of disease outcomes pre-infection.

Method - Statistical Analysis: Please explain what is meant by "the first three ICD diagnoses." There may be a discrepancy in understanding of the first X number of ICD diagnoses in HES between the authors and the reviewer.

Response: We thank the reviewer for these suggestions. We made it clear by adding: "Fourth, we restricted the identification of outcomes only using the first three ICD diagnoses (*out of a series of ICD codes up to 10*), which indicates the main causes for each hospital admission."

Method - Statistical Analysis: Exploratory analysis of non-infected individuals may not be appropriate. Firstly, it's irrelevant to the study question. Secondly, while these individuals have no infection records, it's likely they were not assessed/recorded rather than being uninfected.

Response: We thank the reviewer for these suggestions. As healthy lifestyle is associated with lower risk of chronic

diseases and mortality also among general population before pandemic, we conduct exploratory analysis using a parallel comparison setting to compare the effects of healthy lifestyle on adverse outcomes following COVID-19 with the effects among participants without infection. We observed that lifestyle has stronger effects on multisystem sequelae in the infected group during the acute phases (0-30d following infection), suggesting the particular benefits of healthy lifestyle in preventing acute outcomes that could be directly and biologically caused by viral infection, which strengthen the main findings. We agreed with the reviewer that several participants in the uninfected group may have undiagnosed/untested/asymptomatic COVID-19. We acknowledge this in the limitation: “*Seventh, it's important to acknowledge that some participants classified as uninfected may have had undiagnosed or untested COVID-19. However, by linking participants to official national databases for COVID-19 testing and hospitalization, the likelihood of misclassifying infected and uninfected participants was minimized.*”

Results: Please provide a flowchart.

Response: We thank the reviewer for these suggestions. We now add a flowchart that illustrate study design, cohort construction, and timeline (**Supplementary Fig. 2**).

Reviewer #4 (Remarks on code availability):

I did not recognise that the code was available.

Response: We thank the reviewer for these suggestions. We add the **Code Availability** part and the Analysis code used for this study will be deposited in a GitHub repository at the time of publication.

Reference

- 1 Firth, J. *et al.* A meta - review of “lifestyle psychiatry” : the role of exercise, smoking, diet and sleep in the prevention and treatment of mental disorders. *World psychiatry* **19**, 360-380 (2020).
- 2 Hamer, M., Kivimäki, M., Gale, C. R. & Batty, G. D. Lifestyle risk factors, inflammatory mechanisms, and COVID-19 hospitalization: A community-based cohort study of 387,109 adults in UK. *Brain, behavior, and immunity* **87**, 184-187 (2020).
- 3 Sharif, K. *et al.* Physical activity and autoimmune diseases: Get moving and manage the disease. *Autoimmunity reviews* **17**, 53-72 (2018).
- 4 Tsampasian, V. *et al.* Cardiovascular disease as part of Long COVID: A systematic review. *Eur J Prev Cardiol*, doi:10.1093/eurjpc/zwae070 (2024).
- 5 Gregson, J. *et al.* Cardiovascular risk factors associated with venous thromboembolism. *JAMA cardiology* **4**, 163-173 (2019).
- 6 Davis, H. E., McCorkell, L., Vogel, J. M. & Topol, E. J. Long COVID: major findings, mechanisms and recommendations. *Nature Reviews Microbiology* **21**, 133-146 (2023).
- 7 Ding, D., Rogers, K., van der Ploeg, H., Stamatakis, E. & Bauman, A. E. Traditional and Emerging Lifestyle Risk Behaviors and All-Cause Mortality in Middle-Aged and Older Adults: Evidence from a Large Population-Based Australian Cohort. *PLoS Med* **12**, e1001917, doi:10.1371/journal.pmed.1001917 (2015).
- 8 Batty, G. D., Gale, C. R., Kivimäki, M., Deary, I. J. & Bell, S. Comparison of risk factor associations in UK Biobank against representative, general population based studies with conventional response rates: prospective cohort study and individual participant meta-analysis. *BMJ (Clinical Research ed.)* **368**, m131, doi:10.1136/bmj.m131 (2020).
- 9 Allen, N. E. *et al.* Prospective study design and data analysis in UK Biobank. *Sci Transl Med* **16**, eadf4428, doi:10.1126/scitranslmed.adf4428 (2024).
- 10 Sabia, S. *et al.* Association between questionnaire- and accelerometer-assessed physical activity: the role of sociodemographic factors. *Am J Epidemiol* **179**, 781-790, doi:10.1093/aje/kwt330 (2014).
- 11 Dashti, H. S. *et al.* Genome-wide association study identifies genetic loci for self-reported habitual sleep duration supported by accelerometer-derived estimates. *Nature Communications* **10**, 1100, doi:10.1038/s41467-019-

08917-4 (2019).

- 12 Doherty, A. *et al.* Large Scale Population Assessment of Physical Activity Using Wrist Worn Accelerometers: The UK Biobank Study. *PLoS One* **12**, e0169649, doi:10.1371/journal.pone.0169649 (2017).
- 13 Tsampasian, V. *et al.* Risk factors associated with Post- COVID-19 condition: a systematic review and meta-analysis. *JAMA Internal Medicine* (2023).
- 14 Wang, S. *et al.* Adherence to healthy lifestyle prior to infection and risk of post-COVID-19 condition. *JAMA Internal Medicine* **183**, 232-241 (2023).
- 15 Bryazka, D. *et al.* Population-level risks of alcohol consumption by amount, geography, age, sex, and year: a systematic analysis for the Global Burden of Disease Study 2020. *The Lancet* **400**, 185-235 (2022).
- 16 Lescinsky, H. *et al.* Health effects associated with consumption of unprocessed red meat: a Burden of Proof study. *Nature Medicine* **28**, 2075-2082, doi:10.1038/s41591-022-01968-z (2022).
- 17 Caleyachetty, R. *et al.* United Kingdom Biobank (UK Biobank): JACC Focus Seminar 6/8. *J Am Coll Cardiol* **78**, 56-65, doi:10.1016/j.jacc.2021.03.342 (2021).

REVIEWERS' COMMENTS

Reviewer #1 (Remarks to the Author):

The authors have adequately addressed my comments. Thank you.

Reviewer #1 (Remarks on code availability):

The code has not been shared - the authors stated they will share upon acceptance

Reviewer #2 (Remarks to the Author):

Reviewer #4 (Remarks to the Author):

No comment.